# MoSEL: Modular Self-Reflective Learning for Embodied Decision-Making

## Abstract

Enabling robots to autonomously perform complex, long-horizon tasks remains challenging due to the need for hierarchical reasoning and dynamic adaptability. Humans overcome this by interacting with environment and learning from their own experience, which is infeasible for existing robots without human supervision. To enable similar capabilities in robotic agents, we introduce MoSEL, an modular self-reflective learning framework for robotic decision making. MoSEL combines hierarchical planning with multimodal foundation models, including LVLMs, video diffusion, and inverse dynamics models. These components work together to break down complex tasks, generate executable visual plans, and perform actions. We further introduce a modular self-reflective learning framework that autonomously identifies failures and iteratively refines policies with minimal human intervention. Evaluations on LIBERO-LONG and RoboTwin benchmarks demonstrate that MoSEL outperforms existing methods, achieving over $33\%$ and $46\%$ average performance improvements, respectively. Our results underscore the effectiveness of autonomous self-improvement and accurate failure identification in advancing robust robotic manipulation.

## 1 Introduction

Enabling robots to autonomously perform complex, long-horizon tasks remains a significant challenge in unseen scenario. Such tasks often involve multiple intermediate steps requiring them to dynamically adapt their actions as the environment changes. For instance, assembling furniture need to recognize parts, grasp appropriate tools, and connect components or secure screws. To accomplish these tasks effectively, robots must decompose high-level goals into actionable subtasks and continuously refine their policies through experience.

Hierarchical methods (Zhu et al., 2021; Bacon et al., 2017) are widely used in robotics to address complex tasks by decomposing them into modular subcomponents, thereby enabling flexible planning through the transformation of intricate goals into simpler, manageable steps. Despite their effectiveness, these approaches often struggle to generalize to new scenarios without extensive human intervention, limiting their practicality in dynamic environments. To improve generalization, foundation models such as Large Vision-Language Models (LVLMs) (Achiam et al., 2023; Team et al., 2023) and Vision-Language Models (VLMs) (Radford et al., 2021) have been applied to robotics, transferring broad prior knowledge from web-scale data to embodied tasks (Kim et al., 2024). Meanwhile, Diffusion and Video Diffusion Models (Rombach et al., 2022; Ho et al., 2022) have emerged as powerful tools for bridging language-based goals and physically executable actions, enhancing interpretability and performance through visual planning (Du et al., 2023). Building on these advances, compositional foundation models (Ajay et al., 2023b) integrate hierarchical planning with foundation model reasoning, enabling robots to pursue long-horizon goals across both spatial and temporal scales. However, bridging distribution gaps during module composition remains challenging, often leading to poor adaptability in novel planning scenarios, insufficient physical grounding, and difficulties in correcting execution errors. As a result, these models typically require additional task-specific data or human intervention for finetuning, limiting their effectiveness in dynamic, real-world environments.

To address the limitations of compositional foundation models and enable self-improvement with minimal human intervention, we propose a framework that emulates human-like learning through an

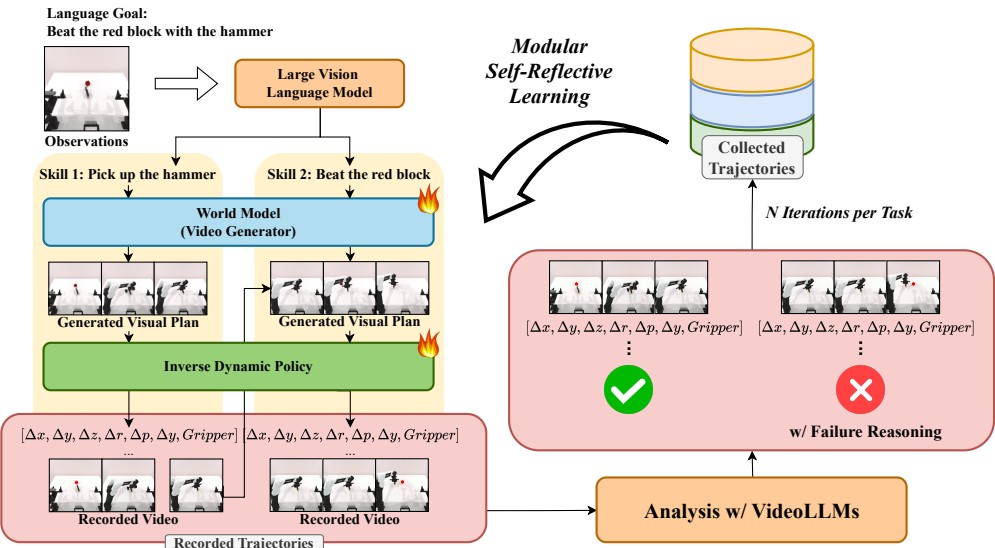

Figure 1: **MoSEL framework overview**. In each trajectory, we identify the cause of failure modes and leverage this information to iteratively improve compositional foundation models for robotic decision-making.

iterative process of trial and error—exploring diverse solutions and learning from both successes and failures. Central to our approach is a modular self-reflective mechanism that autonomously identifies, evaluates (Liu et al., 2023c), and incorporates interaction data from robot-environment engagements. By leveraging advanced Video Large-Language Models (Achiam et al., 2023; Team et al., 2023) to analyze recorded interactions, this mechanism facilitates the continual refinement of compositional components, significantly reducing reliance on human supervision and improving performance in novel long-horizon tasks.

Our key contributions include:

- A novel modular self-reflective learning framework that integrates compositional and foundation model-based approaches to enable robotic self-improvement through interaction-driven learning.
- An autonomous learning framework that leverages advanced VideoLLM to analyze robot-environment interaction videos, identify performance gaps, and refine compositional policy components without human intervention.
- Empirical validation demonstrating significant improvements in long-horizon task success rates across diverse and previously unseen environments.

## 2 RELATED WORK

Prior work on compositional generation of robotic policies has extensively explored the decomposition of complex tasks into modular subcomponents, facilitating efficient and flexible long-horizon planning (Yang et al., 2023). Hierarchical methods commonly leverage modular architectures where high-level task planners generate symbolic or language-based subgoals (Huang et al., 2022; Liang et al., 2023; Lin et al., 2023), and conditional policy networks convert these subgoals into executable low-level actions (Chi et al., 2023; Mete et al., 2024; Janner et al., 2022; Ajay et al., 2023a; Wang et al., 2023b; Urain et al., 2023). Similarly, approaches that maintain libraries of reusable skills or motion primitives, known as skill planners, systematically combine these primitives to handle intricate, multi-stage tasks (Ahn et al., 2022; Zhang et al., 2023). Collectively, these methods underscore the importance of modular policy generation in addressing the complexities inherent in real-world robotic manipulation tasks.

Building upon these compositional approaches, foundation models have emerged as powerful tools for decision making. Particularly, pretrained LVLMs (Achiam et al., 2023; Team et al., 2023; Cla, 2024; Alayrac et al., 2022; Liu et al., 2023b) and VLMs (Shridhar et al., 2022; Wang et al., 2023a) have demonstrated remarkable capabilities in enhancing planning and generalization. Large-scale pretrained models, including LLM-based backbones (Brohan et al., 2022; 2023; Kim et al.,

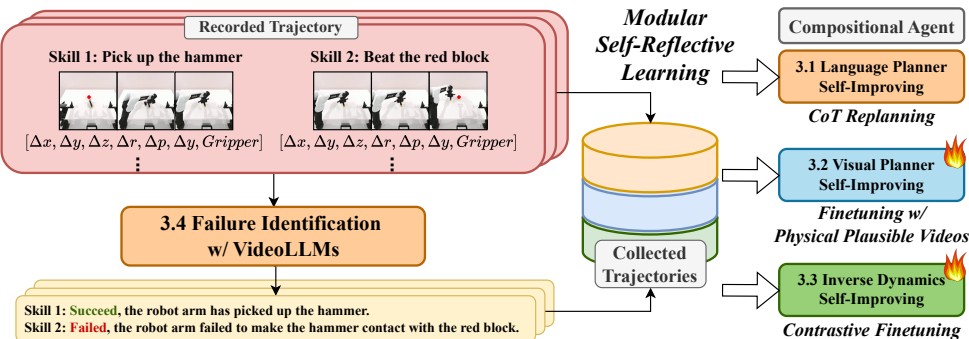

Figure 2: **Modular self-reflective learning pipeline.** The core of our self-improving approach lies in accurately identifying failure modes. To achieve this, we propose a pipeline that evaluates collected interaction trajectories using VideoLLMs.

2024; Team et al., 2025; Zhao et al., 2023) and diffusion transformer models (Liu et al., 2025), effectively transfer vision-language prior knowledge to embodied tasks. These models excel at decomposing high-level instructions into detailed actionable steps through zero-shot or few-shot prompting, requiring minimal task-specific data (Driess et al., 2023; Li et al., 2022; Zeng et al., 2023) and enabling robust generalization across diverse scenarios. Notably, video diffusion models (Huang et al., 2025; Ho et al., 2022; Hong et al., 2023; Jiang et al., 2024) bridge the gap between high-level language goals and executable robotic actions by generating detailed visual plans (Ajay et al., 2023b; Du et al., 2023; Ko et al., 2024; Luo & Du, 2025; McCarthy et al., 2024). The integration of broad prior knowledge embedded within these foundation models has led to significant advancements in the scalability and adaptability of robotic decision-making systems.

Extending these advancements, we propose a framework that strengthens compositional foundation models for hierarchical decision-making by integrating structured reasoning with video large-language models (Achiam et al., 2023; Cla, 2024; Team et al., 2023; Huang et al., 2024b; Wang et al., 2024; Ren et al., 2024; Huang et al., 2024a; Qian et al., 2024). MoSEL introduces a self-strengthening mechanism that autonomously learns from self-generated interactions, enabling continual performance improvement with minimal human supervision.

## 3 METHOD

We propose MoSEL, an modular self-reflective learning framework which is capable of decomposing an unseen long-horizon goal into hierarchical plans and self-improving with the advantage of large video language models. Following prior work (Ajay et al., 2023b) we introduced a compositional foundation model that decomposes long-horizon language-guided tasks into three hierarchical levels:

  **i) Task Planning**: Using a pretrained LVLM to decompose a language goal $g$ into skills $\{w_i\}$ conditioned on initial observation $o_{1,1}$.

  **ii) Visual Planning**: Generating physically plausible observation trajectories $\tau_{i,o} = \{o_{i,1:T}\}$ for each skill $w_i$ using a video diffusion model $\epsilon_\phi$.

  **iii) Action Planning**: Inferring action trajectories $\tau_{a,i} = \{a_{i,1:T-1}\}$ from observation trajectories using an inverse dynamics model $p_\psi$.

The model factorizes the distribution over plans as:

$$p_\Theta(W, \{\tau_{o,i}\}, \{\tau_{a,i}\}|g, o_{1,1}) = \underbrace{\left(\prod_{i=1}^{H} p_\theta(w_i|g, o_{1,1})\right)}_{\text{task planning}} \underbrace{\left(\prod_{i=1}^{H} p_\phi(\tau_{o,i}|w_i, o_{i,1})\right)}_{\text{visual planning}} \underbrace{\left(\prod_{i=1}^{H} \prod_{t=1}^{H-1} p_\psi(a_{i,t}|o_{i,t}, o_{i,t+1})\right)}_{\text{action planning}} \quad (1)$$

where $H$ denotes the action horizon—the number of skills to be executed. This hierarchical approach forms the foundation of our current work. To maximize the joint distribution in Eq. (1) without human supervision, we further introduce an additional component, the **(iv) Video Analyzer** $f_\theta$, which leverages self-collected observations $\{\tilde{\tau}_{o,i}\}_{i=1}^{H}$ gathered during interaction. Figure 2 presents an overview of our framework. The corresponding pseudocode for our modular self-reflective learning procedure is provided in Algorithm 1.

## 3.1 LANGUAGE PLANNING VIA LARGE VISION-LANGUAGE MODELS

Given a language-specified goal $g$ and an initial observation $o_{1,1}$, we use a pretrained multimodal large language model (LVLM) $p_\theta$ as a task planner to generate a sequence of high-level actions $\{w_i\}_{i=1}^H$, where $H$ is the action horizon. This sequence, drawn from the distribution $p_\theta(\{w_i\}_{i=1}^H \mid g, o_{1,1})$, decomposes the goal into skills. By leveraging semantic priors from large-scale image and language pretraining, the LVLM captures procedural knowledge relevant to $g$. To align the planner's output with downstream policies presented in Eq. (1), we constrain action sampling to a finite skill set $S$ of low-level primitives, following (Ahn et al., 2022). However, even state-of-the-art LVLMs struggle to produce optimal plans aligned with the joint distribution in Eq. (1), particularly for complex, long-horizon tasks involving tool manipulation. We need to further optimize the task planner.

**Language Planner Self-Improving**. We introduce a procedure that maximizes the joint distribution in Eq. (1) by incorporating feedback obtained from video analyzer $f_\theta$. We first sample an initial language plan $\{w_i^{(0)}\}_{i=1}^H \sim p_\theta(\{w_i\}_{i=1}^H \mid g, o_{1,1})$, then we execute the high-level language plan in the environment by visual planning and action planning as detailed in Eq. (1). This procedure yields an observation sequence $\{\tilde{\tau}_{o,i}^{(0)}\}_{i=1}^H$, which is the recorded video of execution. The video analyzer $f_\theta$ then generates a rationale $r^{(0)} = f_\theta(\{\tilde{\tau}_{o,i}^{(0)}\}_{i=1}^H)$ in natural language given the recorded video $\{\tilde{\tau}_{o,i}^{(0)}\}_{i=1}^H$. Next we sample a refined plan $\{w_i^{(1)}\}_{i=1}^H \sim p_\theta(\{w_i\}_{i=1}^H \mid g, o_{1,1}, r^{(0)})$ conditioned on this rationale. The overall procedure can be formulated as:

$$\{w_i^{(k+1)}\}_{i=1}^H = \arg \max_{\{w_i\}_{i=1}^H \subseteq S} \log p_\theta(\{w_i\}_{i=1}^H \mid g, o_{1,1}, r^{(k)}).$$

Iteratively applying this process enables the language planner to correct semantic errors without retraining, improving alignment with both environmental constraints and the learned action policy.

## 3.2 VISUAL PLANNING WITH VIDEO GENERATION

Given a language skill $w_i$ from the task planning step, the visual planner generates a sequence of future observations $\tau_{o,i} = \{o_{i,t}\}_{t=1}^T$ conditioned on the initial observation $o_{i,1}$ and skill $w_i$. The visual planner is represented as a video diffusion model parameterized by $p_\phi(\tau_{o,i} \mid w_i, o_{i,1})$. To incorporate task-specific motion priors into the video diffusion model, we first pretrain the visual planner on a dataset of expert demonstrations, $\mathcal{D}_{\text{expert}} = \{(\tau_{o,i}, w_i)\}$, comprising physically plausible trajectories corresponding to each skill:

$$\phi = \arg \max_\phi \ \mathbb{E}_{(\tau_{o,i}, w_i) \sim \mathcal{D}_{\text{expert}}} [\log p_\phi(\tau_{o,i} \mid w_i, o_{i,1})].$$

Despite pretraining, visual plans generated by the diffusion model can still fail in real-world execution due to unrealistic, occlusion or ambiguous motions.

**Self-Improving Stage:** To strengthen visual plans for greater feasibility and consistency over plans, we perform finetuning with $N$ successful observed rollouts $\{\{\tilde{\tau}_{o,i}^{(k)}\}_{i=1}^H\}_{k=1}^N$. Specifically, we collect some recorded videos $\{\tilde{\tau}_{o,i}\}_{i=1}^H$ through interaction with environment as detailed in Eq. (1). We then filter them with video analyzer $f_\theta$, and retrain only $N$ successful videos as a new dataset $\mathcal{D}_{\text{success}} = \{(\{\tilde{\tau}_{o,i}\}_1^H, g)\}$. The finetune procedure can be formulated as:

$$\phi^* = \arg \max_\phi \ \mathbb{E}_{(\{\tilde{\tau}_{o,i}\}_{i=1}^H, g) \sim \mathcal{D}_{\text{success}}} [\log p_\phi(\{\tau_{o,i}\}_{i=1}^H \mid g, o_{1,1})].$$

This procedure ensures that the visual planner produces trajectories that are both semantically consistent and physically plausible. The final visual plan is sampled as $\{\tau_{o,i}\}_{i=1}^H \sim p_{\phi^*}(\{\tau_{o,i}\}_{i=1}^H \mid g, o_{1,1})$, where the model parameters $\phi^*$ have been updated to better align visual imagination with previous successful experiences, thereby enhancing robustness without additional human supervision or reward modeling.

## 3.3 ACTION PLANNING WITH INVERSE DYNAMICS

After obtaining a physically plausible visual plan $\tau_{o,i}$, we utilize an inverse dynamics model to generate an action $a_{i,t}$ conditioned on the current observation $o_{i,t}$ and predicted visual frame $o_{i,t+1}$ following (Pathak et al., 2018). We parameterize this process with language skill guidance $w_i$ as:

$$a_{i,t} \sim p_\psi(a_{i,t} \mid o_{i,t}, o_{i,t+1}, w_i).$$

---

**Algorithm 1:** Modular Self-Reflective Learning for Decision Making

---

**Require:** Task Planner $p_\theta$, Video Diffusion Model $\epsilon_\phi$, Inverse Dynamic $p_\psi$, Video Analyzer $f_\theta$

**Input:** Initial Observation $o_{1,1}$, Long-horizon Task Goal $g$

**Hyperparameters:** Interacting Iteration $N$, Refining Target

1  # Interacting with environment;

2 **for** $n \leftarrow 1$ **to** $N$ **do**

3    Generate language plan $\{w_i\}_{i=1}^H \sim p_\theta(w_i \mid g, o_{1,1})$;

4    **for** $i \leftarrow 1$ **to** $H$ **do**

5       Generate visual plan $\tau_{o,i} \sim p_\phi(w_i \mid g, o_{1,1})$;

6       **for** $t \leftarrow 1$ **to** *skill horizon* **do**

7          Execute with inverse dynamics $a_{i,t} \sim p_\psi(a_{i,t} \mid o_{i,t}, o_{i,t+1}, w_i)$;

8          Collect trajectories $\tau_i \leftarrow \{\tilde{o}_t, a_t\}$;

9    # Failure identification performed by VideoLLM $f_\theta$;

10   $\mathcal{D}_{\text{buffer}} \leftarrow \{w_i, \tau_{a,i}, \tilde{\tau}_{o,i}, f_i, r_i\}_{i=1}^H, \quad f_i \in \{0, 1\}$

11 # Modular Self-Improving;

12 **if** *Language plan self-improving* **then**

13   $r^{(0)} = f_\theta(\{\tilde{\tau}_{o,i}^{(0)}\}_{i=1}^H); \{w_i^{(1)}\}_{i=1}^H \sim p_\theta(\{w_i\}_{i=1}^H \mid g, o_{1,1}, r^{(0)})$;

14 **else if** *Visual plan self-improving* **then**

15   $\phi^* = \arg\max_\phi \ \mathbb{E}_{(\{\tilde{\tau}_{o,i}\}_{i=1}^H, g) \sim \mathcal{D}_{\text{success}}} \left[\log p_\phi(\{\tau_{o,i}\}_{i=1}^H \mid g, o_{1,1})\right]$;

16 **else if** *Inverse dynamics self-improving* **then**

17   $\psi^* = \arg\max_\psi \mathbb{E}_{\mathcal{D}_{\text{success}}}[\log p_\psi(a_{i,t} \mid o_{i,t}, o_{i,t+1}, w_i)] - \mathbb{E}_{\mathcal{D}_{\text{fail}}}[\log p_\psi(a_{i,t} \mid o_{i,t}, o_{i,t+1}, w_i)]$;

18 **return** $\{w_i^{(1)}\}_{i=1}^H \vee \phi^* \vee \psi^*$

---

However, even when pretrained on task-specific low-level skills, the inverse dynamics model occasionally fails to generate feasible actions in scenes with variations. To address this, we aim to enhance the robustness of the inverse dynamics model.

**Inverse Dynamics Self-Improving**. To strengthen the inverse dynamic model, we collect trajectory rollouts $\{w_i, \tilde{\tau}_{o,i}, \tau_{a,i}\}$ consisting of language skill $w_i$, recorded observations $\tilde{\tau}_{o,i} = \{\tilde{o}_{i,1:T}\}$ and action plans $\tau_{a,i} = \{a_{i,1:T-1}\}$, from the interactions with environment similar to the previous approach. We also categorize them into successful and failed executions with $f_\theta$. We then apply contrastive learning to update our action policy, leveraging both successful and unsuccessful trajectories to optimize action discrimination. Specifically, we define the collected datasets as $\mathcal{D}_{\text{success}} = \{(a_{i,t}, \tilde{o}_{i,t}, \tilde{o}_{i,t+1}, w_i)\}$ for successful trajectories and $\mathcal{D}_{\text{fail}} = \{(a_{i,t}, \tilde{o}_{i,t}, \tilde{o}_{i,t+1}, w_i)\}$ for unsuccessful trajectories.

The action policy is optimized using a contrastive objective that simultaneously maximizes the log-likelihood of successful actions and minimizes that of failed actions:

$$\psi^* = \arg\max_\psi \mathbb{E}_{\mathcal{D}_{\text{success}}} \left[\log p_\psi(a_{i,t} \mid o_{i,t}, o_{i,t+1}, w_i)\right] - \mathbb{E}_{\mathcal{D}_{\text{fail}}} \left[\log p_\psi(a_{i,t} \mid o_{i,t}, o_{i,t+1}, w_i)\right].$$

This procedure advances the model's ability to distinguish executable and robust actions from those likely leading to failures, thus significantly improving execution reliability and task success.

### 3.4 IDENTIFYING FAILURE WITH VIDEO LARGE-LANGUAGE MODELS

We outline the video analysis process performed by $f_\theta$ in Algorithm 1, comprising the following two steps: (a) **Sub-goal evaluation** – determining a sequence of sub-goals necessary to achieve a given skill sequence $\{w_i\}_{i=1}^H$ and assessing whether each sub-goal has been successfully completed. (b) **Failure identification** – employing advanced video-language models to diagnose specific reasons for failure observed in the execution video $\{\tilde{\tau}_{o,i}\}_{i=1}^H$.

**Sub-goal Evaluation**. We identify sub-goals for each skill to evaluate achievement at each stage. Given a sequence of predicted low-level skills $\{w_i\}_{i=1}^H$ with action horizon $H$ and an initial environment observation $o_{1,1}$, we prompt a LVLM to convert each skill into a sub-goal. For example, a skill "pick up the hammer" converts to the sub-goal "The gripper holds the hammer", while "Beat

| Kitchen | Living Room | Studying Room | Brush | Hammer | Brush & Dustpan |

(a) LIBERO Benchmark (b) RoboTwin Benchmark

Figure 3: **Benchmarks** We rigorously evaluate MOSEL on two robotic manipulation benchmarks within simulation environments. For the LIBERO benchmark, we focus on long-horizon, multi-scene scenarios, while for RoboTwin, our emphasis is on long-horizon tool-use tasks.

the red block with hammer" becomes "The hammer head has contacted the red block." Using the recorded video captured during environment interaction, we then prompt the VideoLLM to assess whether each sub-goal has been achieved. If all the sub-goals have been accomplished, we regard the execution consisting of $H$ skill trajectories as successful and add it to the reflection buffer. If any of the sub-goals has not been fulfilled, we need to further identify the specific reason.

**Failure Identification with VideoLLMs**. Given a failed video trajectory $\{\tilde{\tau}_{o,i}\}_1^H$ from the previous stage, we first prompt the VideoLLM to identify the reason for failure. If the failure is attributed to the language plan, we update the task planner as described in Sec. 3.1. When the generated language plan is deemed reasonable, we then use the video analyzer $f_\theta$ to determine which specific skill caused the failure. Our objective is to collect such reflection data into a buffer $\mathcal{D}_{\text{buffer}}$ as follows:

$$\mathcal{D}_{\text{buffer}} \leftarrow \{w_i, \tau_{a,i}, \tilde{\tau}_{o,i}, f_i, r_i\}_{i=1}^H, \quad f_i \in \{0, 1\}$$

where $f_i = 0$ denotes a failed skill trajectory, $f_i = 1$ denotes a successful skill trajectory and $r_i$ denotes the corresponding reasoning.

This skill-wise evaluation enables us to precisely label skill trajectories, self-generating valuable training data to further advance our compositional agent's performance on specific sub-tasks. As illustrated in Figure 2, the trajectory of first skill "pick up the hammer" will be labeled as a succeed sequence, while the trajectory sequence of second skill "Beat the red block with hammer" will be labeled as a failed sequence because the agent fails to move the hammer to the red block precisely.

## 4 EXPERIMENTS

We evaluate MOSEL on long-horizon planning tasks drawn from diverse distributions, including varying object positions and combinations. Our approach is compared against existing compositional planning methods. We further analyze the impact of the self-improving framework at each hierarchical level and investigate how iterative self-reflective learning enables task specialization with minimal human supervision or predefined environmental rewards.

### 4.1 BASELINES

We compare our approach with several existing approaches construct robot manipulation policies conditioned on language goals using compositional models:

**Video Planner**. We compare our approach with a video diffusion model (UniPi) (Du et al., 2023) $\{\tau_o^i\} \sim p(\{\tau_o^i \mid o_{i,t}, g\})$ without task planning, generates video plans for the entire task and generates actions $a_{i,t}$ using an inverse dynamics model $a_{i,t} \sim p(\{a_{i,t} \mid o_{i,t}, o_{i,t+1}\})$.

**Language Planner**. We compare our approach with a hierarchical system (PaLM-E) (Ahn et al., 2022; Driess et al., 2023) with LLM/LVLM as high level task planner that sequences skills from a set of learned skills to accomplish a long-horizon task. We utilize GPT-4o (Achiam et al., 2023) as task planner and train our low-level policy (Mete et al., 2024; Chi et al., 2023) $\{a_{i,t:T-1}\} \sim p(\{a_{i,t:T-1} \mid o_{i,t}, w_i\})$ conditioned on the current observation and language goal.

**Language and Video Planner**. We compare our approach with a system that combines both language planner and visual planner as detailed in Eq. (1). We use it as our baseline approach and construct it following the HiP framework (Ajay et al., 2023b), integrating a LVLM, video diffusion model, and inverse dynamics model, but excluding the iterative self-reflective learning mechanism from their work, which requires additional training of task-specific classifiers.

|  | Task1 | Task2 | Task3 | Task4 | Task5 | Task6 | Task7 |
|---|---|---|---|---|---|---|---|
| UniPi (Du et al., 2023) | 0.0 | 0.0 | 0.0 | 0.0 | 0.30 | 0.0 | 0.0 |
| PaLM-E (Driess et al., 2023) | 0.06 | 0.28 | 0.80 | 0.04 | 0.0 | 0.10 | 0.16 |
| MOSEL (Baseline) | 0.48 | 0.40 | 0.44 | 0.10 | 0.26 | 0.34 | 0.32 |
| +Self-Improved Video | 0.58 | 0.48 | 0.66 | 0.06 | 0.36 | 0.32 | 0.40 |
| ++Self-Improved Policy | **0.86** | **0.88** | **0.90** | **0.42** | **0.86** | 0.49 | **0.75** |
| †VLA-RL (Lu et al., 2025) | 0.58 | 0.64 | 0.44 | 0.34 | 0.82 | **0.50** | 0.66 |

Table 1: MOSEL outperforms all hierarchical planning baselines on unseen long-horizon tasks across seven settings, which collectively span three distinct scenes in the LIBERO-LONG benchmark. We omit the number of task planner self-advancing in this evaluation, as the language goals are relatively simple and do not pose significant challenges for our planner. († We provide additional comparison with RL-based approach, for more details please refer to the supplementary.)

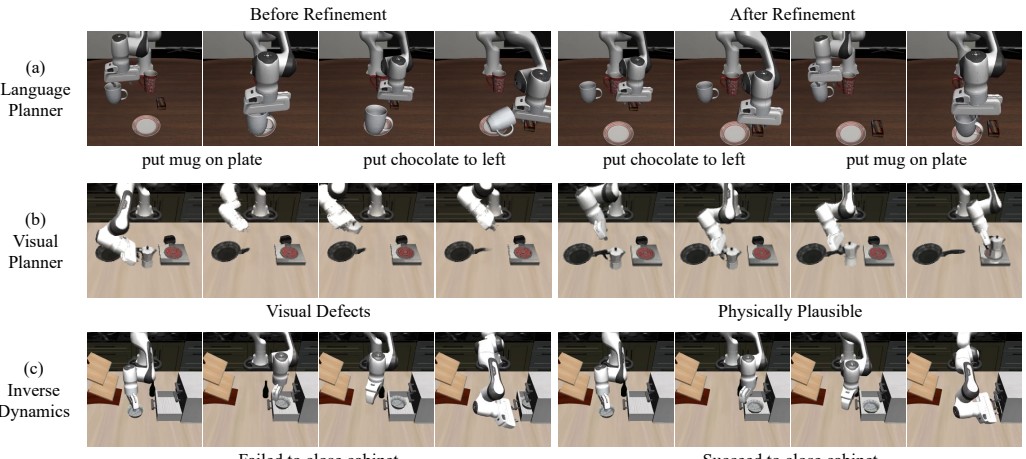

Figure 4: **Qualitative Visualization of LIBERO** We present examples illustrating the self-improving process. (a) demonstrates how the generated task plan can be restructured. (b) shows how visual defects in the generated video can be corrected.(c) details how the action policy can self-improve to ensure correct execution.

## 4.2 EXPERIMENTS ON LIBERO

We first evaluate MOSEL on the LIBERO-LONG set from LIBERO (Liu et al., 2023a) benchmark, which tests the ability to combine multiple learned skills in new situations with different object combinations and positions. This presents a challenge for transferring skills to long-horizon tasks. Following our self-reflective learning procedure, results in Table 1 show that the MOSEL framework significantly improves performance on all tasks in the LIBERO-LONG benchmark, achieving an average performance gain of 33% on seven long-horizon tasks compared to the baseline approach. To further illustrate the effectiveness of our approach, we highlight three representative tasks in Fig. 4, each demonstrating our method's robustness in distinct failure scenarios.

**Language Planner Self-Improving**. In Fig. 4-(a) left, the task goal is "Put the white mug on the plate and put the chocolate pudding to the right of the plate." The initial language plan sequenced these actions accordingly. However, video analysis revealed that during execution, the end effector collided with the mug in the second step, causing failure. After analyzing this failure, the refined plan reversed the action sequence: "Put the chocolate pudding to the right of the plate. / Put the white mug on the plate." This example demonstrates how action ordering can significantly impact execution success. The improved results using our framework can be seen in Fig. 4-(a) right.

**Visual Planner Self-Improving**. In Fig. 4-(b) left, the originally generated visual plan contained defects that made it physically implausible. After refining the world model, the improved visual plan shown in Fig. 4-(b) right displays clearer frames and more reasonable transitions.

**Inverse Dynamics Self-Improving**. In Fig. 4-(c), the robot initially failed to properly close the cabinet drawer. Following action policy self-improving, as illustrated in Fig. 4-(c) right, the robot demonstrated stable execution of control actions.

| Methods | (a) Sweep-Block | (b) Push-Block | (c) Beat-Block |
|---|---|---|---|
| UniPi (Du et al., 2023) | 0.0 | 0.0 | 0.2 |
| PaLM-E (Driess et al., 2023) | 0.02 | 0.20 | 0.16 |
| MoSEL (Baseline) | 0.0 | 0.10 | 0.20 |
| +Self-Improved LVLM | 0.44 | 0.40 | 0.32 |
| ++Self-Improved Video | 0.46 | 0.44 | 0.40 |
| + + +Self-Improved Policy | **0.52** | **0.58** | **0.60** |

Table 2: MoSEL outperforms all hierarchical planning baselines in the RoboTwin tool-use tasks. Our experiments reveal that refining the same components across scenarios leads to varying degrees of improvement, highlighting the importance of accurately identifying failures and selectively advancing components.

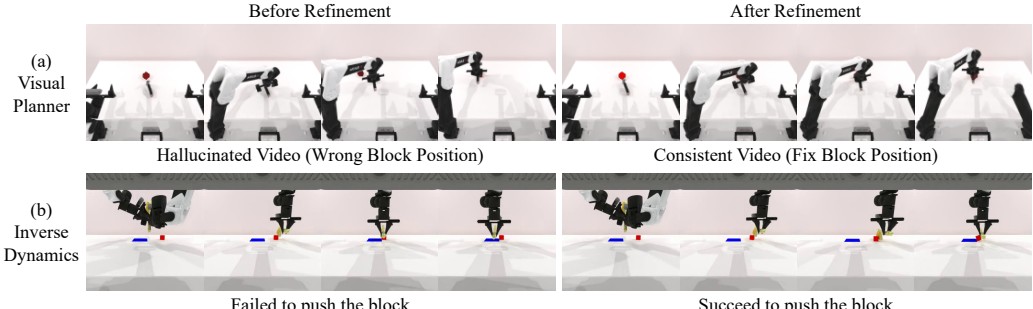

Figure 5: **Qualitative Visualization of RoboTwin** We present a case in which the hallucinated visual plan guides the arm to an incorrect position, as shown in (a)-left. In contrast, although the generated visual plans in (b)-left are accurate, the inverse dynamics fails to execute the actions with sufficient accuracy and precision. The improvement shown on the right illustrates the result after refining the video diffusion and inverse dynamics.

### 4.3 Experiments on RoboTwin

We next evaluate MoSEL on the RoboTwin (Mu et al., 2025) benchmark, which focuses on tool manipulation and requires the planner to sequence tool-use steps using a dual-arm robot. A key challenge is choosing the correct arm to reach and operate on target objects. For instance, in the *Beat-Block* task, the red target block may be positioned such that only one arm can access it, making the correct sequencing of tool usage critical for task success. Following the self-reflective learning procedure, as shown in Table 2, the MoSEL framework demonstrates substantial improvement on all tasks in the RoboTwin benchmark, achieving an average performance gain of 46.7%. We further illustrate the effectiveness of our approach through three representative failure scenarios:

**Task Planner Self-Improving**. We found that generating a feasible language plan could be a difficult task in a complex interaction scenario. Specifically, for tool-using tasks such as *Push-Block* and *Sweep-Block*, how to generate a language plan to correctly manipulate the tools by a dual-arm robot requires a lot of physical knowledge and needs to be aware of the cause-effect relationships within actions (Chen et al., 2024). In *Sweep-Block* task, the goal is "brush the red block into the dustpan" and the robot needs to pick up the brush with left hand and handover it to the right. If we naively prompt the task planner to generate a sequence of actions, it often neglects the process of "handover the brush". After the self-improving procedure, the VideoLLM could detect the failure reasons and include "handover object" in the next generation. We provide the re-generating and reasoning process in the supplementary.

**Visual Planner Self-Improving**. We also encounter scenarios where compositional models generate incorrect visual plans. Specifically, information loss or occluded objects can lead to erroneous transitions within visual plans. As illustrated in Fig. 5 (a), in the *Beat-Block*, the target becomes occluded by the robotic arm itself after tool pickup. Conditioned only on the current observation, the world model incorrectly predicts the target block's position, causing the policy to direct the robotic arm to an incorrect location shown in the visual plan. To mitigate this failure mode, incorporating previously observed information proves crucial. Our self-improving procedure enables diffusion model to acquire knowledge of physically plausible transitions between skills. This approach improves success rates on this task from 32% to 40% as shown in Table 2 after self-improvement.

**Inverse Dynamics Self-Improving**. Another common failure occurs when multitask policies are inadequately trained for specific tasks, leading to incorrect execution. In the *Push-Block* task, the

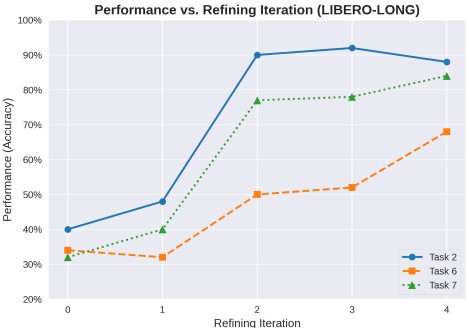

Figure 6: **Iteratively Self-Improving** This figure illustrates that, with MOSEL, the agent system can continuously self-improve on a specific task.

|  | Language Acc(%) | Visual Acc(%) |
|---|---|---|
| w/o MOSEL | 26.0 | 55.3 |
| w/ MOSEL | 89.3 | 92.7 |

Table 3: **Improvements of Intermediates** Quantitative improvements observed at intermediate stages to highlight the effectiveness of the self-improving process. († Language plan accuracy is based on ground-truth comparison; visual accuracy is determined by human verifiers)

robot needs to push a red block to a blue target region using a tool. The generated language and video plans are typically correct. However, observed from a different angle, we find that the policy merely imitates the motion without achieving the goal of moving the target to its destination as shown in Fig. 5 (b) left. After the self-improving procedure on inverse dynamics we see a significant improvement on this task from $44\%$ to $58\%$ success rate.

## 4.4 ANALYSIS

We conduct experiments showing that, with MOSEL, the model can be continually advanced on a specific task. Moreover, we present improvements in intermediate outputs, including both language and visual plans. Finally, we highlight the critical role of the failure identification procedure.

**Iterative Self-Improving**. We conducted experiments on the LIBERO-LONG benchmark to evaluate our framework's ability to iteratively improve task performance. Across four iterations on three tasks—alternating between world model (iterations 1 and 3) and action policy self-improving (iterations 2 and 4)—we observed steady performance gains Fig. 6. Action policy updates yielded the most substantial improvements, while world model self-improving had diminishing returns after the initial iteration.

| Acc(%) | Task 2 | Task 3 | Task 5 |
|---|---|---|---|
| Baseline | 0.48 | 0.66 | 0.36 |
| w/o FI | 0.48 | 0.56 | 0.22 |
| w/ FI | **0.90** | **0.90** | **0.86** |

Table 4: **Effectiveness of Failure Identification** The results indicate that incorporating failure identification (FI) into self-collected interactions is integral to the success of the strengthening procedure.

**Enhancements to Intermediate Representations**. We present the enhanced intermediate representations (language plan and visual plan) following self-improving procedures, as detailed in Table 3. To illustrate the improvements in language planning, we select three tasks from RoboTwin, chosen for their inherent difficulty in task planning. Additionally, we select three tasks from LIBERO to demonstrate the effectiveness of visual quality enhancement. These examples collectively provide evidence for the crucial role of improving intermediate modules in boosting the overall success rate.

**Ablation on Failure Identification**. We compare the effects of including versus omitting failure identification during the action plan self-improving procedure. In this experiment, we remove the failure identification step and instead directly collect all interaction trajectories to finetune the inverse dynamics model. Experiments are conducted on three tasks sampled from LIBERO-LONG, with results summarized in Table 4. The findings indicate that distinguishing between successful and failed self-collected samples significantly improves the effectiveness of the self-improving procedure.

## 5 CONCLUSION

Our research introduces a novel framework that addresses the challenge of complex task execution in robotics through modular self-reflective learning. By integrating hierarchical decomposition with foundation models, our approach enables autonomous trial-and-error learning with minimal human intervention. This integration advances the adaptability of robotic systems, bridging the gap between high-level AI models and real-world physical capabilities, and paving the way for continuous self-improvement in dynamic environments.

## 6 ETHICS STATEMENT

The proposed framework has the potential to change the field of robotics by allowing robots to learn and adapt to complex tasks with very little human intervention. By using compositional generative policies and self-reflective learning, our approach reduces the need for expert guidance and helps robots work well in changing and unpredictable environments. This progress could be very useful for areas such as healthcare, eldercare, manufacturing, where robots can help solve labor shortages, increase productivity, and improve safety. Making this kind of technology more accessible could also help smaller organizations and less privileged communities, supporting greater social equality.

## 7 REPRODUCIBILITY STATEMENT

We describe the main framework in Sec. 3. Detailed implementation information, including prompts, experimental settings, computational resources, and algorithms, is provided in the supplementary material. In addition, we release essential code through an anonymized GitHub link, specifically covering the components that utilize VideoLLM for failure identification and language-plan replanning. These efforts are intended to ensure the reproducibility of our work.

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

## SUPPLEMENTARY MATERIAL

## A  ANONYMOUS PROJECT PAGE

For more visualization of the strengthening results, please refer to  recap2025.github.io

## B  CODE

For the detailed codes and prompts of the failure identification as described in Sec. 3.4, please refer to  Anonymous GitHub

## C  LIMITATIONS

Our system is built on advanced proprietary multimodal large language models that have video understanding abilities, which help to distinguish and evaluate different types of interaction data. However, we noticed that the system sometimes hallucinates, especially in some scenarios, making it difficult to correctly tell apart successful and failed trajectories. These problems mainly come from the current vision-language models' limitations, especially when they need to handle unclear or new visual inputs in a novel environment. To deal with this issue, we currently rely on human intervention through prompt engineering in our framework. Achieving a fully automatic system that can improve itself without human help is still something we hope to explore in the future. Also, while our framework strengthens compositional foundation models for specific tasks, this focus can reduce their original ability to handle multiple tasks at once. This shows a basic challenge in AI: there is always a trade-off between optimizing for one task and keeping the model's general skills. One possible way to overcome these limitations is to introduce lifelong learning or continual learning. With lifelong learning, the model could learn new skills with little data, keep what it learned before, and avoid forgetting its original functions.

However, as robots become more autonomous, there are also new challenges. These include issues with transparency, accountability, and ethics, especially since robots may develop in unexpected ways. There is also a risk that the introduction of advanced robots could affect job markets and increase inequality if only certain groups have access to them. Therefore, it is important to ensure the safety and reliability of these self-learning systems, which will require regulation and open discussion.

## D  LLM USAGE STATEMENT

We employed LLMs in three contexts: (1) serve as one of the key components (language planner) in our designed architecture (2) serve as one of the key components (Video Analyzer) in our self-reflection mechanism; and (3) to assist in refining the grammar and semantics of the manuscript. The scientific contributions and claims of this work are solely those of the authors, and we take full responsibility for all content.

## E  SIMULATION ENVIRONMENTS

**LIBERO** (Liu et al., 2023a). LIBERO is a benchmark for lifelong learning in robot manipulation that focuses on how agents acquire and transfer both declarative knowledge (about entities) and procedural knowledge (skills and actions) over time. The benchmark includes 130 diverse tasks and human demonstration data to support imitation learning. We pretrain our video diffusion model and inverse dynamics policy on the 90 low-level skills from LIBERO-90 set, and evaluate our system on seven tasks from the LIBERO-LONG set, which require multi-step executions to achieve the specified goals.

**RoboTwin** (Mu et al., 2025). RoboTwin provides a generative digital twin framework for dual-arm robotic manipulation that addresses the lack of diverse, high-quality data in this domain. It leverages 3D generative models and large language models to create realistic objects from single images, generate interactive scenarios, and produce spatially-aware control code. In our evaluation, we use

two tasks from the original benchmark and design one additional task. All three tasks as illustrated in main paper Fig. 3 involve tool manipulation and require the planner to reason through the sequential steps of tool usage to complete the assigned objectives.

## F    EXPERIMENTAL DETAILS

**Train and Test Setting**.  For both environments, we curate 50 demonstrations per skill to train the low-level policies. During evaluation, we use a distinct set of random seeds to ensure that the arrangement of target objects differs from those seen during training. We assess the success rate of the complete long-horizon task by conducting 50 trials, each with a unique seed per task. Notably, the full long-horizon tasks are also unseen during training. For the strengthening process, we collect $50 - 100$ interactions per round to obtain a sufficient number of successful demonstrations. This facilitates more stable finetuning of both the video diffusion models and the inverse dynamics model.

**Base Models**.  We construct our HiP (Ajay et al., 2023b) like compositional foundation models with the following three modules: (a) task planner, we utilize GPT-4o (Achiam et al., 2023) as task planner with strong prior image-language knowledge; (b) visual world model, we pretrain a Flowdiffusion (Ko et al., 2024) as our video generator; (c) inverse dynamics models, we utilize diffusion policy (Chi et al., 2023) as a low-level controller for LIBERO environment; we utilize a vector-quantized transformer (Mete et al., 2024) as a low-level policy for RoboTwin environment. Both architectures are trained with future image conditioning, where the target image is randomly sampled within a predefined 32-step window from the same trajectory. For the video analyzer $f_\theta$, we use GPT-4.1 (OPENAI, 2025) and Gemini-Pro-2.5 (DeepMind, 2025), featuring with strong video understanding capability. Particularly, we found that Gemini-Pro-2.5 serves as a very powerful failure detector.

**Details for LIBERO**. For the pretraining, we train a multimasks policy with all 50 demonstrations in each of the 90 tasks in the LIBER-90. For the strengthening procedure, we collect 100 interactions in the simulation. We further finetune the visual world model or inverse dynamics model with only the collected data in each round of strengthening procedure. In Table 5, we provide the tasks we select to evaluate in main paper Tab. 2.

| Task | Task Descriptions |
|---|---|
| Task 1 | Put both the alphabet soup and the tomato sauce in the basket. |
| Task 2 | Turn on the stove and put the moka pot on it. |
| Task 3 | Put the black bowl in the bottom drawer of the cabinet and close it. |
| Task 4 | Put the white mug on the left plate and put the yellow and white mug on the right plate. |
| Task 5 | Pick up the book and place it in the back compartment of the caddy. |
| Task 6 | Put the white mug on the plate and put the chocolate pudding to the right of the plate. |
| Task 7 | Put both the alphabet soup and the cream cheese box in the basket. |

Table 5: Tasks selected from LIBERO-LONG

**Details for RoboTwin**. For the pretraining, we train a multimasks policy with 50 demonstrations generated by code (Mu et al., 2025) in each of the 24 skills in the RoboTwin Benchmark as demonstrated in Listing 1. For strengthening procedure, we first collect 50 interactions in the simulation. We further finetune the visual world model or inverse dynamics model with both the collected data and pretraining trajectories in each round of strengthening procedure. We retrieve the trajectories of skills used in the predicted language plan from the pretraining data in the finetuning stage to reduce the time cost of interaction process. In Table 6 we provide the tasks we select to evaluate in main paper Tab. 3.

| Task | Task Descriptions |
|---|---|
| Beat-Block | Beat the red block with the hammer. |
| Push-Block | Push the red block to the target blue region with the brush. |
| Sweep-Block | Sweep the red block to the dustpan with the brush. |

Table 6: Tasks selected from RoboTwin

**Comparing with RL-based Approach**. We include an empirical evaluation against VLA-RL (Lu et al., 2025), a recent method that uses RL-based finetuning with ground-truth rewards. We fixed the total number of online improvement trials to ensure a fair comparison. Notably, ReCap achieves superior performance without requiring predefined reward signals, highlighting its self-improvement capability as detailed in Tab. 1. These results support a key observation: RL-based finetuning with a monolithic policy (VLA-RL) transfers poorly to long-horizon tasks under limited interaction (about 100 per tasks). Task 5, which requires only one skill, is the only one where VLA-RL gets relatively higher score among all tasks, underscoring the advantage of our proposed modular self-reflective learning framework.

## G    ADDITIONAL EXPERIMENTAL RESULTS

We provide some additional experiments to demonstrate the robustness and effectiveness of our proposed method MOSEL. Due to the limited computational resources, we conduct the following experiments on a selected set of LIBERO-Long benchmark.

### G.1    RESISTANCE TO VISUAL DISTURBANCES

To assess the robustness our method, and to simulate the impact of pixel-to-action errors, we further conduct experiments with visual disturbances on observations with the following condition change:

- Brightness Adjustment with factor sampled from [0.8, 1.2]
- Contrast Adjustment with factor sampled from [0.9, 1.1]
- Color Temperature Shift with red/blue gain smapled from [0.95, 1.05]
- Gamma Correction to mimic camera exposure and sensor response with power factor sampled from [0.9, 1.1]

|  | Task 2 | Task 6 | Task 7 |
|---|---|---|---|
| Without visual disturbances | 0.88 | 0.49 | 0.75 |
| With visual disturbances | 0.94 | 0.50 | 0.80 |

Table 7: Performance comparison with and without visual disturbances.

These results suggest that our policy is robust to visual shifts, further supporting its potential for sim-to-real transfer.

### G.2    FAILURE-IDENTIFICATION ACCURACY

We report the failure-identification accuracy of VideoLLM (GPT-4.1) and its impact on downstream task success after one round of refinement. The results show that precision is a key factor influencing final performance.

High precision ensures that the model correctly identifies true succeed causes, enabling effective and targeted refinements. In contrast, low precision leads to many false positives, causing the refinement stage to address incorrect issues and degrade performance.

For instance, as depicted in Table 8, Task 6 shows lower precision (0.64) and a reduced success rate after refinement (0.49), despite high recall. This suggests that over-identification harms effectiveness. Conversely, Task 2 achieves high precision (0.89) and the highest improvement (0.88), confirming that accurate pinpointing of failure sources directly correlates with successful recovery.

We further provide the failure-identification performance of a state-of-the-art open-source model, Qwen-2.5-72B-Instruct (Bai et al., 2025), as shown in Table 9 below. This model was evaluated across three key tasks in our study.

Despite its strong general language capabilities, Qwen-2.5-72B-Instruct performs poorly when applied as a video verifier in the robotic self-improvement loop, especially in precision—failing to

| | Video Analysis Result | | | | Robot Accuracy |
|---|---|---|---|---|---|
| Task | Acc | Precision | Recall | F1 Score | Success Rate After Refinement |
| Task 2 | 0.82 | **0.89** | 0.62 | 0.73 | **0.88** |
| Task 6 | 0.78 | **0.64** | 0.82 | 0.72 | **0.49** |
| Task 7 | 0.83 | **0.85** | 0.55 | 0.67 | **0.75** |

Table 8: Comparison between video analysis metrics and robot accuracy across tasks.

| Task | Acc | Precision | Recall | F1 Score |
|---|---|---|---|---|
| Task 2 | 0.72 | **0.64** | 0.91 | 0.75 |
| Task 6 | 0.60 | **0.00** | 0.00 | 0.00 |
| Task 7 | 0.76 | **0.00** | 0.00 | 0.00 |

Table 9: Video analysis metrics across tasks.

identify correct behaviors in Tasks 6 and 7. In contrast, proprietary VideoLLMs such as GPT-4.1 and Gemini consistently provide more accurate and balanced evaluations without any task-specific finetuning.

These results highlight two key points:

- Current open-source LLMs/VLMs remain insufficient for high-precision video-based evaluation in the context of robotic learning. As a result, it is difficult to integrate existing open-source LLMs/VLMs into an overall self-improvement loop.

- Our work is the first to successfully demonstrate how advanced proprietary VideoLLMs can be used as plug-and-play verifiers, enabling self-improvement in compositional robotic agents without finetuning.

We hope our findings can motivate future research toward improving the video understanding capabilities of open-source models.

### G.3 STATISTICAL CONFIDENCE REPORTING

We provide the reproduced results for the three selected tasks. Despite the randomized elements in the strengthening process, the results consistently demonstrate performance improvements, highlighting the robustness and effectiveness of our method.

| | Task 2 | Task 6 | Task 7 |
|---|---|---|---|
| RECAP (Baseline) | $0.395 \pm 0.005$ | $0.315 \pm 0.025$ | $0.32 \pm 0.00$ |
| +Strengthen Video | $0.43 \pm 0.05$ | $0.37 \pm 0.05$ | $0.43 \pm 0.03$ |
| ++Strengthen Policy | $0.88 \pm 0.02$ | $0.49 \pm 0.01$ | $0.75 \pm 0.05$ |

Table 10: Task performance with baseline and strengthened methods.

### G.4 ABLATION STUDY ON SAMPLING STEPS

We conduct an ablation study on one task to evaluate how this reduction affects both performance and execution time. The results below report task accuracy and total execution time for 50 one-by-one trials, along with the average per-video sampling time:

| De-noising Steps | Acc | Execution Time | Per Video Sampling Time |
|---|---|---|---|
| 50 | 0.88 | 60 min | 31 sec |
| 20 | 0.82 | 43 min | 12 sec |
| 10 | 0.82 | 37 min | 5 sec |

Table 11: Impact of de-noising steps on accuracy and execution time.

## H COMPUTATIONAL RESOURCES

We conduct our experiments on two resources: GeForce RTX 3090 GPU and 8 V100 GPUs. We use 3090 GPU for evaluation and pretraining, finetuning the inverse dynamic models. We use the 8 V100 GPUs for diffusion model pretraining and finetuning.

Listing 1: Low-level skills of RoboTwin

```
[
    'open the cabinet with left hand',
    'open the cabinet with right hand',
    'pick up the apple with left hand',
    'pick up the apple with right hand',
    'put the apple in the cabinet with left hand',
    'put the apple in the cabinet with right hand',
    'close the cabinet with left hand',
    'close the cabinet with right hand',
    'pick up the red block with left hand',
    'hand over the red block',
    'put the red block on the blue target region with right hand',
    'pick up the hammer with left hand',
    'pick up the hammer with right hand',
    'beat the red block with the hammer with left hand',
    'beat the red block with the hammer with right hand',
    'pick up the brush with left hand',
    'hand over the brush',
    'push the red block to the blue target region with the brush with
        right hand',
    'pick up the dustpan with left hand',
    'sweep the red block into the dustpan with the brush',
    'pick up the mug with left hand',
    'place the mug in front of the rack',
    'pick up the mug with right hand',
    'hang the mug on the rack',
]
```

## I FINETUNING INVERSE DYNAMICS MODEL WITH CONTRASTIVE LEARNING

We provide the approach about how we finetune the inverse dynamics model with both the succeed and failed samples as detailed in Sec. 3.3. Our approach uses contrastive learning (Chen et al., 2020) to enhance the action policy. Using collected trajectories with both successful and failed cases, we define $S$ as the set of indices for positive samples and $F$ for negative samples, with $\ell(\hat{a}, a)$ as the base loss function between predicted action $\hat{a}$ and target action $a$. The loss for the positive samples is defined as:

$$L_{\text{succeed}} = \sum_{i \in S} \ell(\hat{a}_i, a_i)$$

For the negative samples, with a margin $m$, the loss is defined as:

$$L_{\text{failed}} = \sum_{j \in F} \max\left\{0, m - \left|\ell(\hat{a}_j, a_j)\right|\right\}$$

the overall loss function is then given by:

$$L = L_{\text{succeed}} + \alpha \, L_{\text{failed}}$$

where $m$ and $\alpha$ are hyperparameters that prevent loss explosion or divergence. This approach increases the probability of positive action distribution while decreasing the negative action distribution.

## J  SKILL TRANSITION

We encountered a challenge when applying a policy pretrained on low-level skills to long-horizon tasks: the robot remained at the final pose after completing each skill, since all demonstrations began from a fixed initial pose and ended at the task's target pose. Consequently, the policy failed to learn appropriate transitions between skills. While one solution is to collect new expert demonstrations that start from random initial poses, this approach is labor-intensive. Instead, we opt to reset the arm to its original pose after each skill execution. Specifically, we record the initial end-effector pose at the start of each task and use a pose controller to return the arm to this configuration before initiating the next skill. In our controller, for the first $N_1$ steps, we command upward motion to the end-effector to avoid potential collisions, after which we execute full pose correction to the initial configuration. The procedure is detailed as follows:

Let $\boldsymbol{p}_{\text{init}} \in \mathbb{R}^3$ and $\boldsymbol{p}_{\text{cur}} \in \mathbb{R}^3$ denote the initial and current end-effector positions, and $\boldsymbol{q}_{\text{init}}, \boldsymbol{q}_{\text{cur}} \in \mathbb{R}^4$ denote the corresponding quaternions for orientation.

The position and orientation errors are computed as:

$$\Delta \boldsymbol{p} = \boldsymbol{p}_{\text{init}} - \boldsymbol{p}_{\text{cur}}$$
$$\boldsymbol{q}_{\text{rel}} = \boldsymbol{q}_{\text{init}} \otimes \boldsymbol{q}_{\text{cur}}^{-1}$$
$$\Delta \boldsymbol{o} = \text{QuatToAxisAngle}(\boldsymbol{q}_{\text{rel}})$$

Define the raw action vector:

$$\boldsymbol{a}_{\text{raw}} = \begin{bmatrix} \Delta \boldsymbol{p} \\ \Delta \boldsymbol{o} \end{bmatrix} \in \mathbb{R}^6$$

Normalize the action by the maximum allowed values:

$$\boldsymbol{s}_{\text{max}} = [0.05, \, 0.05, \, 0.05, \, 0.5, \, 0.5, \, 0.5]$$

$$\boldsymbol{a}_{\text{norm}} = \text{clip}\left( \frac{\boldsymbol{a}_{\text{raw}}}{\boldsymbol{s}_{\text{max}}}, \, -1, \, 1 \right)$$

The final action $\boldsymbol{a}$ at each time step is:

$$\boldsymbol{a} = \begin{cases} [0, \, 0, \, \boldsymbol{a}_{\text{norm},z}/5, \, 0, \, 0, \, 0, \, -1] & \text{if step} \leq N_1 \\ [\boldsymbol{a}_{\text{norm}}, \, -1] & \text{otherwise} \end{cases}$$

where $N_1$ is a predefined step.

## K  FAILURE CASES

A common failure mode arises when the compositional agent fails to execute one of the skills in the generated long-horizon plan. In such cases, it becomes impossible to collect a successful demonstration for the strengthening procedure. We observe this type of failure in three remaining LIBERO-LONG tasks: "put both the cream cheese box and the butter in the basket", "put both moka pots on the stove", and "put the yellow and white mug in the microwave and close it". These observations highlight the system's dependence on a generalist and robust low-level policy to reliably execute skills.

Another limitation involves the need for prompt engineering during video analysis. Occlusions or missing information can lead to hallucinated outputs or incorrect predictions. For instance, we must clarify that the "alphabet soup" refers to the can with green and red coloring. This necessity motivates the inclusion of an "<ADDITIONAL HINTS>" placeholder in our system prompts.

## L ADDITIONAL VISUALIZATION OF REFINEMENT

Here we provide more visualizations of the improvement after the strengthening procedure.

### L.1 STRENGTHENING THE TASK PLANNER

We provide three examples (two from RoboTwin and one from LIBERO-LONG) to demonstrate the results of strengthening the task planner.

**Sweep-Block**. In this example, as shown in Fig. 7, a robot is assigned the task of sweeping a red block into a dustpan using a brush. The original language plan instructed the robot to pick up the brush and dustpan sequentially with the left hand before performing the sweeping. However, the execution failed because the robot attempted to grasp both tools simultaneously with the same hand, making it impossible to perform the final task. A revised plan was generated, introducing a handover step to free up the left hand for proper tool handling. This adjustment allowed for sequential tool manipulation and set up the robot to complete the task more effectively.

**Push-Block**. In this example, as shown in Fig. 8, the robot was instructed to push a red block into a blue target region using a brush. Initially, it successfully picked up the brush with its left hand but failed to complete the task because it never transferred the brush to the right hand or used it to interact with the block. The original plan assumed the robot could use its right hand without explicitly including a transfer step. Based on this failure, a revised plan was generated: the robot should first pick up the brush with its left hand, then hand it over to the right hand, and finally push the red block with the brush using the right hand. This adjustment aligns better with the robot's available skill set.

**Put the white mug on the plate and put the chocolate pudding to the right of the plate**. In this example, as shown in Fig. 9, the robot was tasked with placing a white mug on a plate and positioning a chocolate pudding to the right of the plate. The initial execution succeeded in placing the mug but failed overall when the robot's arm knocked the mug over while trying to place the pudding, disrupting the setup. Moreover, the pudding was not successfully placed before the video ended. To address this, a revised language plan was proposed that reverses the action order—placing the pudding first, then the mug—to reduce the chance of collision and ensure successful task completion.

**Task Goal: Sweep the red block into the dustpan with the brush.** User

**Generated Language Plan:** [
'pick up the brush with left hand',
'pick up the dustpan with left hand',
'sweep the red block into the dustpan with the brush'
]

**Analyze the following video to identify the robot's actions ...**

```
{
  "explanation": "
    1. The main goal ('sweep the red block into the dustpan with the brush') can be divided into the following sub-goals:
        (a) pick up the brush with the left hand,
        (b) pick up the dustpan with the left hand, and
        (c) sweep the red block into the dustpan with the brush.

    2. Evaluating fulfillment of each sub-goal:
        - (a) The robot successfully picked up the brush with the left hand in the early frames.
        - (b) The robot then proceeded to pick up the dustpan with the left hand, stacking it with the brush in its grasp,
              instead of picking up the dustpan and brush separately. This does not match the expected sequence.
        - (c) The robot did not proceed to sweep the red block into the dustpan (neither tool was brought near the red block, and no
              sweeping motion occurs).

    3. Action correlation:
        - Sub-goal (a): Correctly fulfilled. The robot's left hand picks up the brush at the start.
        - Sub-goal (b): Partially fulfilled, but incorrectly: The robot tries to pick up the dustpan while already holding the brush, resulting in
              both tools being grasped together but not in a way that would accommodate the sweeping action.
        - Sub-goal (c): Not fulfilled. The robot does not move the tools toward the red block or attempt a sweeping motion.

    4. Reasons for success/failure:
        - Step 1: Successful. Correct grasp of the brush with the left hand.
        - Step 2: Failure/partial. The plan called for sequential pickup (first brush, then dustpan, presumably with another hand or with a
              hand free), but the robot tries to pick up both with the same hand, which is suboptimal and interferes with task execution.
        - Step 3: Failure. Because of the way the tools are held, no attempt is made to sweep the red block, so the primary goal is
              unfulfilled.",

  "task_results": [
    {
      "task_description": "Pick up the brush with left hand.",
      "execution_result": "Succeeded. The robot picks up the brush with the left hand.",
      "task_succeed": true
    },
    {
      "task_description": "Pick up the dustpan with left hand.",
      "execution_result": "Failed. The robot attempts to pick up both the brush (already held) and dustpan together with the left
              hand, rather than separately or as expected.",
      "task_succeed": false
    },
    {
      "task_description": "Sweep the red block into the dustpan with the brush.",
      "execution_result": "Failed. The robot does not move either tool towards the red block or perform a sweeping motion;
              no progress is made toward this goal.",
      "task_succeed": false
    }
  ]
}
```

**Based on the analysis of the video, your task is to regenerate a different language plan.
...** User

```
{
  "explanation": "
    Based on the video and failure analysis, the problem occurred because the robot tried to pick up both the brush and the dustpan
    with the left hand at the same time, which is not supported and does not allow for correct task execution. The first instruction (pick
    up the brush with left hand) succeeded and should be kept. To improve the plan using only the available skill set, the next step
    should be to use the right hand to pick up the dustpan, as 'pick up the dustpan with right hand' is not listed in the skill set, but 'pick
    up the dustpan with left hand' is, and alternatively, the dustpan could be manipulated by handing over the brush (if required by the
    setup). However, since we need both tools to achieve the goal and can sweep directly after picking up the brush, a plausible valid
    sequence from the skills would be:
        (1) pick up the brush with left hand,
        (2) pick up the dustpan with left hand,
        (3) sweep the red block into the dustpan with the brush.
    If the robot can only use the left hand for both pickups, it must put down the brush before picking up the dustpan or use a
    handover. Given only the listed skills, the plan should involve sequential pickup and tool use without holding both simultaneously.
    Since 'pick up the brush with left hand', 'hand over the brush', and 'pick up the dustpan with left hand' exist, the robot may use a
    handover to the right hand to free the left hand for the dustpan pickup. Thus, the steps should be: pick up the brush with left hand,
    hand over the brush, pick up the dustpan with left hand, sweep the red block into the dustpan with the brush.",

  "goal": "Sweep the red block into the dustpan with the brush.",

  "language_plan": [
    "pick up the brush with left hand",
    "hand over the brush",
    "pick up the dustpan with left hand",
    "sweep the red block into the dustpan with the brush"
  ]
}
```

After Strengthening (Recorded Execution)

Figure 7: **Strengthening Task Planner - Case 1.**

**Task Goal: Push the red block to the target blue region with the brush.** | User

**Generated Language Plan: [**
**'pick up the brush with left hand',**
**'push the red block to the blue target region with the brush with right hand'**
**]**

**Analyze the following video to identify the robot's actions ...**

```
{
    "explanation": "
        1. Sub-goals required to achieve the main goal 'push the red block to the blue target region with brush':
            a) Pick up the brush with left hand.
            b) Use the brush (with one of the hands, as needed) to push the red block into the blue target region.

        2. Identify if each sub-goal has been fulfilled:
            a) The robot picked up the brush with its left hand, successfully fulfilling the first sub-goal.
            b) The robot did not push the red block toward the blue target region at any point in the video. The block remained in its place, and there
                was no interaction made between the brush and the red block.

        3. Actions corresponding to each sub-goal:
            a) For picking up the brush with the left hand: The robot moved its left arm to grasp the brush.
            b) For pushing the red block with the brush with the right hand: There is no observed action where either arm pushes the red block using
                the brush. The right hand never takes the brush, and neither hand pushes the red block.

        4. Reasons for success or failure:
            - Instruction 1 ('pick up the brush with left hand'): This was successfully executed, as the robot did pick up the brush with its left hand.
            - Instruction 2 ('push the red block to the blue target region with the brush with right hand'): This failed because the robot never
                transitioned the brush to the right hand nor used either hand to push the red block toward the blue region. The predicted plan was
                not executable after step 1, resulting in the main task being incomplete.",

    "task_results": [
        {
            "skill": "pick up the brush with left hand",
            "execution_result": "The robot successfully picked up the brush with its left hand.",
            "task_succeed": true
        },
        {
            "skill": "push the red block to the blue target region with the brush with right hand",
            "execution_result": "The robot did not push the red block at all; neither was the brush transferred to the right hand or used for pushing.
                                The sub-goal was not fulfilled.",
            "task_succeed": false
        }
    ]
}
```

**Baed on the analysis of the video, your task is to regenerate a different language plan.**
**...** | User

```
{
    "explanation": "
        The failure analysis showed that the robot was able to pick up the brush with the left hand but failed to transfer the brush or use
        it to push the red block. Reviewing the available skill set, the robot cannot explicitly transfer the brush to the right hand, which is
        required for the skill 'push the red block to the blue target region with the brush with right hand.' However, there is a skill 'hand
        over the brush' which can be used to transfer the brush between hands. Thus, the task can be completed by picking up the
        brush with the left hand, handing over the brush, and then pushing the block using the right hand as per the available skill. Here
        is the regenerated language plan using only the skill set skills.",

    "goal": "push the red block to the blue target region with brush",

    "language_plan": [
        "pick up the brush with left hand",
        "hand over the brush",
        "push the red block to the blue target region with the brush with right hand"
    ]
}
```

After Strengthening (Recorded Execution)

Figure 8: **Strengthening Task Planner - Case 2.**

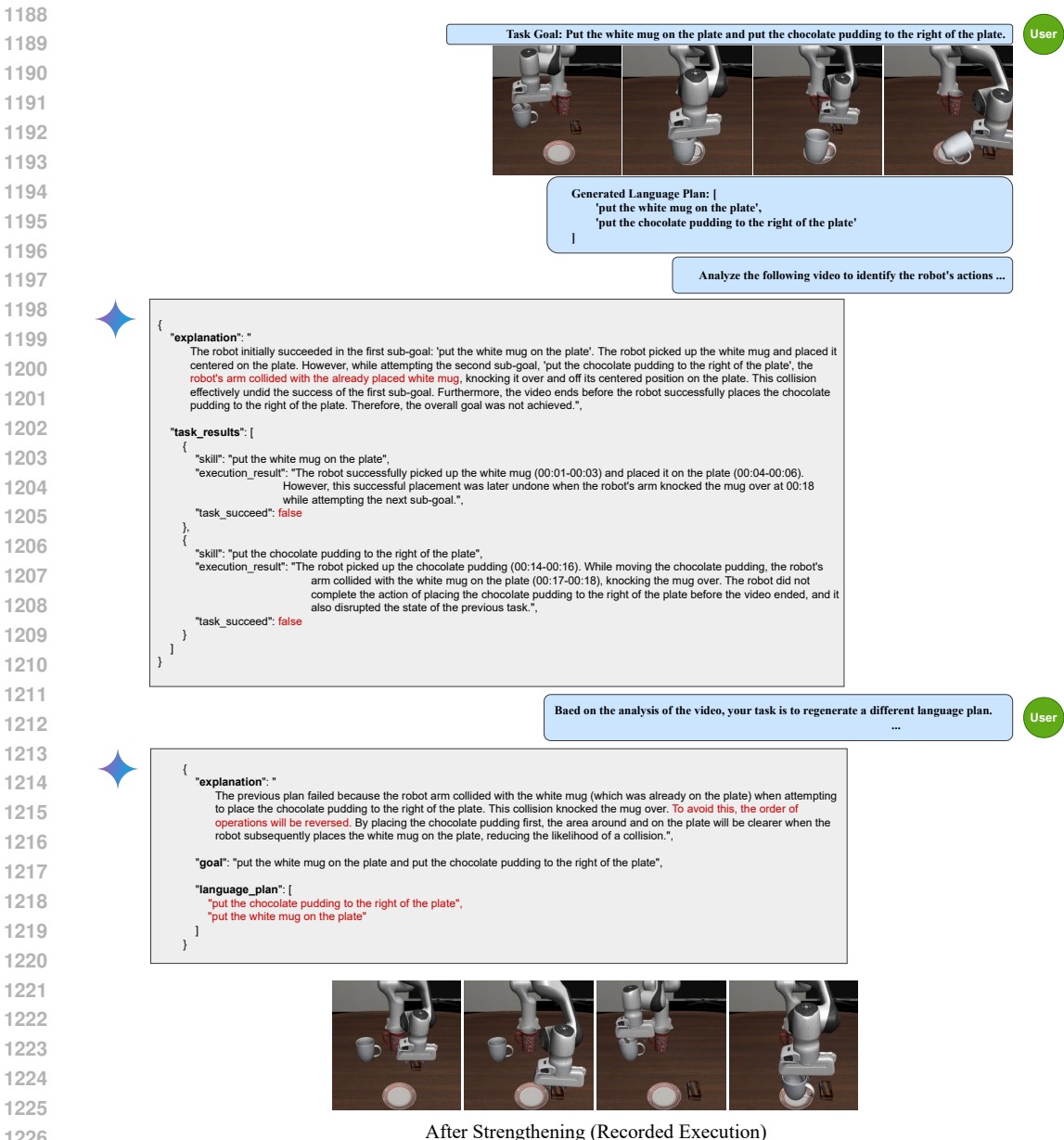

After Strengthening (Recorded Execution)

Figure 9: **Strengthening Task Planner - Case 3.**

## L.2    Strengthening the Visual World Model

We then provide four examples (one from RoboTwin and three from LIBERO-LONG) to demonstrate the results of strengthening the visual world model.

**Beat-Block**. As described in main paper Sec. 4.3, information loss or occluded objects can lead to erroneous transitions within visual plans. As illustrated in Fig. 10 (a), the target becomes occluded by the robotic arm itself after tool pickup. Conditioned only on the current observation as initial image in Fig. 10 (b), the world model incorrectly predicts the target block's position, causing the policy to direct the robotic arm to an incorrect location shown in the visual plan. From Figure 10 (c) and (d) we can see that the robotic arm is guided to a wrongly predicted position. To mitigate this failure mode, incorporating previously observed information proves crucial. Our strengthening procedure enables diffusion model to acquire knowledge of physically plausible transitions between skills. As demonstrated in Fig. 10, the red block stays in the fix position across the whole generated visual plan.

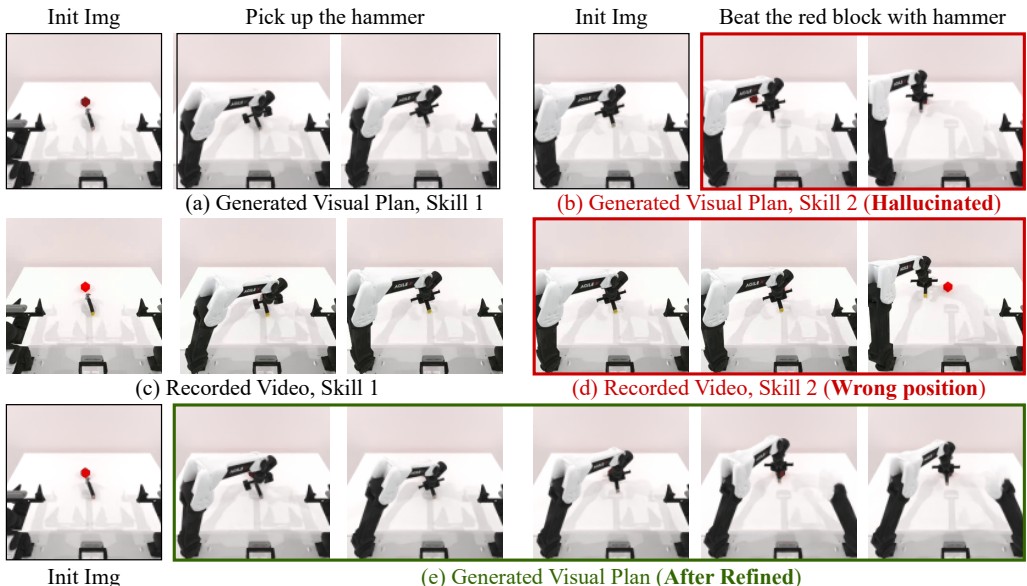

Figure 10: **Strengthening World Modeling - Case 1.** We present another case where object occlusion (b) leads to incorrect visual plan predictions. (d) shows that the inverse dynamics follows the visual plan (b) and move the robotic arm to the wrong position. After refinement, the red block remains consistently positioned across different generated frames (e).

**Pick up the book and place it in the back compartment of the caddy**. In the original generated video Fig. 11 (a), the robot initially moves the book toward the right compartment of the caddy but then abruptly places it in the front compartment, resulting in an implausible motion trajectory. After the strengthening process, as shown in Fig. 11 (b), the robot successfully moves the book to the back compartment with a physically plausible motion.

Generated Videos

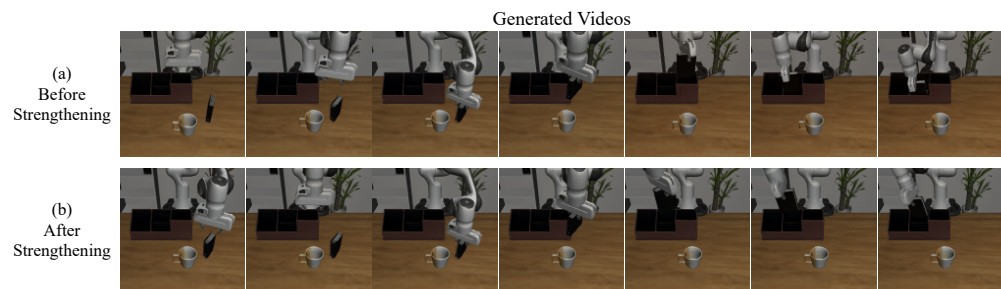

(a) Before Strengthening

(b) After Strengthening

Pick up the book and place it in the back compartment of the caddy.

Figure 11: **Strengthening World Modeling - Case 2.**

**Put the white mug on the plate and put the chocolate pudding to the right of the plate**. In the original generated video Fig. 12 (a), the robot correctly moves the white mug toward the plate but incorrectly moves the chocolate pudding across the mug to the left of the plate, resulting in noticeable visual artifacts. After the strengthening process, as shown in Fig. 12 (b), the robot successfully moves the chocolate pudding to the right of the plate with a physically plausible motion.

Generated Videos

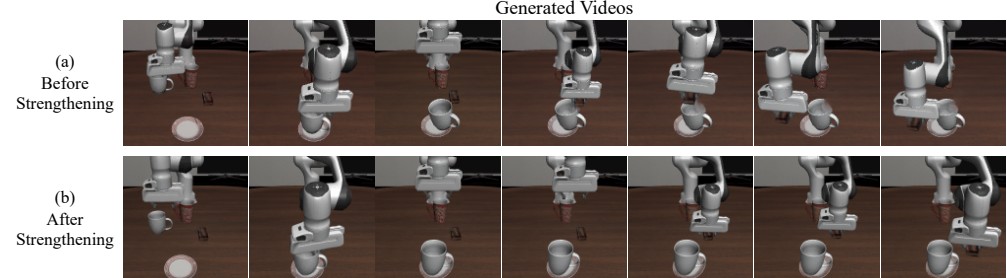

(a) Before Strengthening

(b) After Strengthening

Put the white mug on the plate and put the chocolate pudding to the right of the plate.

Figure 12: **Strengthening World Modeling - Case 3.**

**Put the black bowl in the bottom drawer of the cabinet and close it**. In the original generated video Fig. 13 (a), the robot incorrectly moves the bowl toward the top of the cabinet, and the bowl subsequently disappears in later frames, leading to an implausible object motion. After the strengthening process, as shown in Fig. 13 (b), the robot successfully places the bowl in the bottom drawer of the cabinet and closes it with a physically plausible motion.

Generated Videos

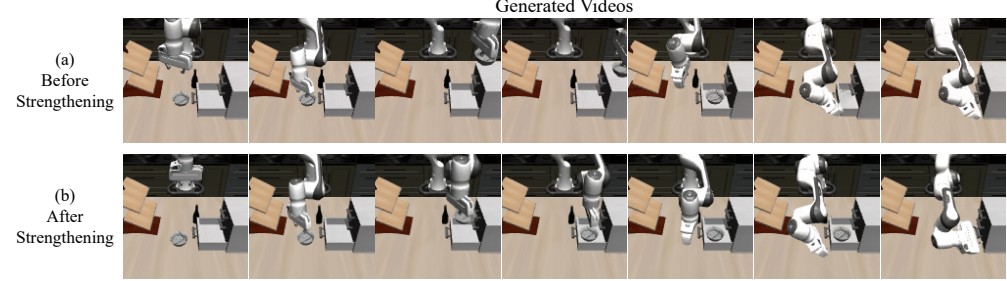

(a) Before Strengthening

(b) After Strengthening

Put the black bowl in the bottom drawer of the cabinet and close it.

Figure 13: **Strengthening World Modeling - Case 4.**

### L.3 STRENGTHENING THE INVERSE DYNAMICS MODEL

We then provide three examples (one from RoboTwin and two from LIBERO-LONG) to demonstrate the results of strengthening the inverse dynamics model.

**Push-Block**. As detailed in main paper Sec. 4.3, even when the inverse dynamics model follows a correct visual plan, as shown in Fig. 14 (a) and (b), the robot may still fail to execute certain precise actions, as illustrated in Fig. 14 (c). To address this, we finetune the inverse dynamics model using the collected data. Following the strengthening procedure, Fig. 14 (d) shows that the robot successfully pushes the red block.

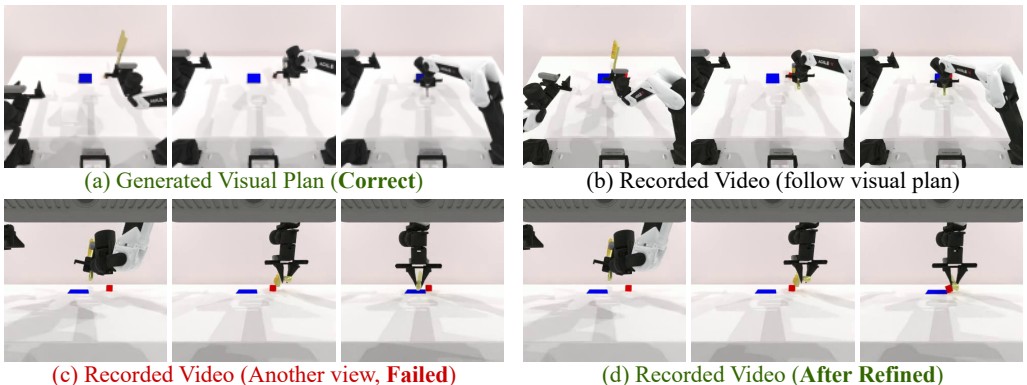

(a) Generated Visual Plan (**Correct**)    (b) Recorded Video (follow visual plan)

(c) Recorded Video (Another view, **Failed**)    (d) Recorded Video (**After Refined**)

Figure 14: **Strengthening Inverse Dynamics - Case 1.** We present another case where the inverse dynamics fails. From (a) we can see that the generated visual plan is generally correct. However, even if the inverse dynamics model follows the visual plan as demonstrated in (b), we can see that the robot fails to push the red block correctly. It only learns to imitate the action. After strengthening procedure, we can see in (d), the inverse dynamics model learns the correct way to operate.

**Put both the alphabet soup and the cream cheese box in the basket**. In the original recorded video Fig. 15 (a), the robot fails to move the alphabet soup toward the cabinet. After the strengthening process, as shown in Fig. 15 (b), the robot successfully moves the the alphabet soup toward the cabinet.

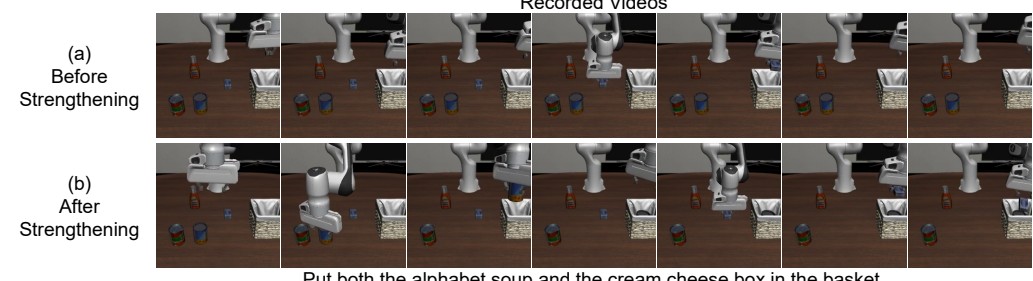

Put both the alphabet soup and the cream cheese box in the basket.

Figure 15: **Strengthening Inverse Dynamics - Case 2.**

**Pick up the book and place it in the back compartment of the caddy**. In the original recorded video Fig. 16 (a), the robot fails to pick up the book. After the strengthening process, as shown in Fig. 16 (b), the robot successfully pick up the book and place it in the target position.

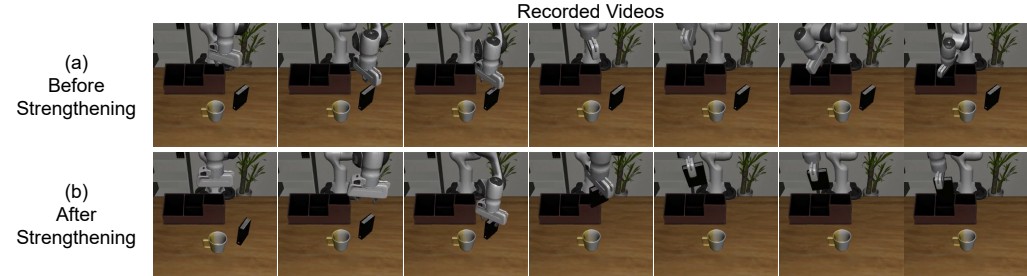

Pick up the book and place it in the back compartment of the caddy.

Figure 16: **Strengthening Inverse Dynamics - Case 3.**

## M    PROMPTS

We provide the prompts we used to generate language plan and failure identification as detailed in main paper Sec. 3.4.

### M.1    PROMPTS FOR GENERATING TASK PLAN

Listing 2: Task Planning Prompt

```
(
    "You are controlling a single-arm robot. Your task is to generate a
        feasible, step-by-step language plan "
    "that maps a language goal to a series of low-level skills (refer to
        LOW_LEVEL_SKILL_SET). "
    "Language goal: " + <TASK DESCRIPTION> + "\n"
    "Follow this general taxonomy:\n\n"
    "1. Environment Analysis: Identify target objects and their locations
         from the observation.\n"
    "2. Target Prioritization: Determine the primary goal and the object(
        s) involved.\n"
    "3. Action Sequencing: Decompose the overall goal into sub-tasks.
        Order actions so that tasks requiring a free hand are scheduled "
    "appropriately.\n"
    f"4. Mapping: For each sub-task, select the corresponding command
        from <SKILL SET>.\n\n"
    "Generate a coherent, sequential plan that adheres to this strategy.\
        n"
    "5. Reasoning: Provide a brief explanation of your plan and the
        rationale behind it.\n"
    "6. Your final output should contain the JSON structure as a string:\
        n"
    "```json\n"
    "{{\"language_plan\": [\"<Skill 1>\", \"<Skill 2>\", \"...\"]}}\n"
    "```"
)
```

## M.2 Prompts for Re-generating Task Plan

Listing 3: Sub-goal Identification and Evaluation

```
Analyze the following video to identify the robot's actions in the video
    are consistent with language goal.
This is a recorded video that the robot performed a task in a simulated
    environment according to the predicted language plan.
However, the robot may not have completed the task successfully.
Sometimes the generated plan may not be correct.
Your task is to determine at which step the robot failed to follow the
    language plan.
The goal is: <GOAL>.
The predicted language plan is: <LANGUAGE_PLAN>.
Provide your explanation following the steps:
1. Find the sub-goals required to achieve the main goal.
2. Identify if each sub-goal has been fulfilled.
3. If any of the sub-goals are not fulfilled, identify the actions in the
    video that correspond to each sub-goal.
4. Identify the reasons for the success or failure of each language
    instruction in the language plan.

Additional hinks:
<ADDITIONAL HINTS>
```

Listing 4: Prompt for Re-planning

```
Baed on the analysis of the video, your task is to regenerate a different
    language plan.
According to failure analysis, the robot may not have completed the task
    successfully.
The goal is: <GOAL>.\n
The predicted language plan is: <LANGUAGE_PLAN>.\n
The history of the video analysis is: <HISTORY>.\n

After analyzing the video, please regenerate the language plan from the
    first failed one.
The language plan should be a list of instructions that the robot should
    follow to complete the task successfully.
```

## M.3 Prompts for Failure Identification

Listing 5: Failure Identification

```
Analyze the following video to identify the actions in the video are
    consistent with the following language plan.
This is a recorded video that the robot performed a task in a simulated
    environment according to the predicted language plan.
However, the robot may not have completed the task successfully.
Your task is to determine if each action in the video corresponds to the
    language plan completely.

The goal is: <GOAL>.
The predicted language plan is: <LANGUAGE_PLAN>.

Provide your explanation following the steps:
1. Find the sub-goals required to achieve the main goal.
2. Identify if each sub-goal has been fulfilled.
3. If any of the sub-goals are not fulfilled, identify the actions in the
    video that correspond to each sub-goal.
4. Identify the reasons for the success or failure of each language
    instruction in the language plan.

Additional hints:
<ADDITIONAL HINTS>
```

