# OpenReview forum: "MoSEL: Modular Self-Reflective Learning for Embodied Decision-Making"
_ICLR.cc/2026/Conference — Submitted to ICLR 2026_

### Official Review · Reviewer_Qf7W · 2025-10-29

**Soundness:** 3
**Presentation:** 3
**Contribution:** 2
**Rating:** 4
**Confidence:** 3

**Summary:**

This paper presents MoSEL (Modular Self-Reflective Learning), a framework for enabling robots to autonomously improve their performance on long-horizon manipulation tasks through iterative self-reflection. Experiments on LIBERO-LONG and RoboTwin benchmarks demonstrate performance improvements of 33% and 46.7% respectively over baseline methods.

**Strengths:**

1. The paper introduces an innovative approach by leveraging VideoLLMs to automatically analyze robot-environment interaction videos and identify failure modes without human supervision. This modular self-reflective mechanism that jointly refines language planning, visual planning, and action planning components represents a creative combination that could reduce the dependency on manual labeling and expert intervention in robotic learning systems.
2. The framework provides a well-structured hierarchical decomposition with clearly defined modules, and demonstrates substantial performance improvements (33-46% gains) on multiple benchmark tasks.

**Weaknesses:**

1.No real-world robot experiments; all evaluations are simulation-only, raising serious questions about practical applicability and sim-to-real transfer.
2.Computational costs (inference time, memory, convergence rate) are completely unanalyzed, making it impossible to assess practical feasibility.
3.Insufficient baseline comparisons with only 3-4 methods; missing recent self-improving and VLM-based approaches; some baselines show 0.0% performance.
4.VideoLLM failure diagnosis accuracy is never quantitatively evaluated; no analysis of false positives/negatives or handling of ambiguous failures.
5.Limited generalization evidence; performance plateaus after 3-4 iterations; no testing on truly out-of-distribution scenarios or novel environments.

**Questions:**

1.Can the authors provide concrete computational costs: inference time per module, total wall-clock time per task, GPU memory requirements, and average iterations to convergence?
2.Why did the authors not include real robot validation, and do the authors have plans for physical experiments to demonstrate sim-to-real transfer?
3.How do the authors explain baseline methods (UniPi, PaLM-E) achieving 0.0% on multiple tasks—are these implementation issues or fair comparisons?
4.What is the precision and recall of VideoLLM failure identification, and have the authors conducted human evaluation of diagnostic accuracy?
5.Have the authors tested on out-of-distribution scenarios with completely novel objects/environments, and can the improvements generalize beyond training variations?

---

> ### Author Response · Authors · 2025-11-18
>
> We sincerely thank the reviewer for the thoughtful and detailed feedback.
>
> **Q1. Can the authors provide concrete computational costs: inference time per module, total wall-clock time per task, GPU memory requirements, and average iterations to convergence?**
>
> We have provided computational requirements (Supplementay H) and inference time (Supplementay G.4). In average, it takes roughly 5-30 seconds per skill for visual planning depending on the de-noising steps (5s for 10 steps and 30s for 50 steps) on a single RTX 3090 GPU.
>
> Regarding the number of iterations required for convergence, we provide detailed iterative refinement experiments in Figure 6 across three representative tasks. Our results reveal that different tasks exhibit distinct convergence patterns, which is an informative finding in itself. This task-dependent convergence behavior highlights a key advantage of our modular self-reflective framework: it enables targeted diagnosis and refinement of individual modules and their intermediate outputs. Unlike end-to-end approaches where failure modes are opaque, our framework allows for interpretable debugging—we can identify whether failures stem from skill selection, world model prediction, or action execution, and refine accordingly. This modularity enhances both the reliability and efficiency of the self-improvement process.
>
> **Q2. Why did the authors not include real robot validation, and do the authors have plans for physical experiments to demonstrate sim-to-real transfer?**
>
> We agree that it would be desirable to have real robot results. We have not conducted real-world experiments due to hardware limitations. That being said, we believe our results are interesting even if they are done in simulation. The major focus of the paper is to construct a modular self-improving system, and the input requirements for each component of our system are similar to those in real robots (our systems operates only on RGB observations and poses) and each step of the self-improvement procedure can be plausibly executed on a real robot. In addition, prior works have further illustrated how a similar stack of models can be reliably executed on physical robots [1]. Finally, simulation gives us a repeatable benchmark in which we can reproducibly assess the overall approach.
>
> **Q3. How do the authors explain baseline methods (UniPi, PaLM-E) achieving 0.0% on multiple tasks—are these implementation issues or fair comparisons?**
>
> UniPi is a visual planning architecture that combines a video diffusion model with an action planner, but lacks a high-level task planner. In our experimental setting, all baselines—including UniPi—only have access to low-level skill demonstrations during training, without exposure to long-horizon task compositions.
>
> Under these constraints, UniPi cannot decompose unseen, long-horizon tasks into executable low-level skills, which explains its 0.0% success rate across multiple tasks. This is not a failure of UniPi per se, but rather demonstrates that flat planning architectures struggle when task decomposition capabilities are not explicitly provided during training.
>
> These results underscore the necessity of our hierarchical structure, which operates across multiple levels of abstraction—semantic (task planning), visual (subgoal prediction), and action (skill execution). This multi-level hierarchy enables effective task decomposition and skill composition even when trained only on isolated low-level skills, a capability that flat architectures cannot achieve under the same data constraints.
>
> In contrast, PaLM-E (language planner + action policy) achieves 0.0% success rate on only one task—Task 5 of LIBERO-Long. Notably, UniPi achieves 30% success on this particular task, highlighting the importance of visual planning for this uniquely challenging scenario. This suggests that certain tasks may benefit more from visual grounding than purely language-based task decomposition.
>
> **Q4. What is the precision and recall of VideoLLM failure identification, and have the authors conducted human evaluation of diagnostic accuracy?**
>
> We have provided detailed experiments for the precision and recall of VideoLLM failure identification in Supplementary G.2, along with table 8 and table 9. Our experiments highlight that high precision ensures that the model correctly identifies true succeed causes, enabling effective and targeted refinements. In contrast, low precision leads to many false positives, causing the refinement stage to address incorrect issues and degrade performance. Additionally, we highlights the shortage of existing open-source multimodal large-language model in table 9. Note that we calculate the performance of failure identification according to environmental ground truth, which we don't access in the refinement stage.

---

> > ### Author Response · Authors · 2025-11-18
> >
> > **Q5. Have the authors tested on out-of-distribution scenarios with completely novel objects/environments, and can the improvements generalize beyond training variations?**
> >
> > Our modular self-reflective learning framework is pretrained on low-level skills and evaluated on **unseen, long-horizon** tasks. For instance, using the LIBERO benchmark, we pretrain our world model and action policy on LIBERO-90 (primarily single-action tasks). We then conduct evaluations on LIBERO-10, which demands **multi-action sequences and introduces novel object and scene arrangements absent from the training set**.
> >
> > While prior work [2,3] has demonstrated that large-scale pretraining can enhance diffusion model generalizability, that approach is distinct from our primary focus on modularity and self-reflective learning. Therefore, such large-scale pretraining is beyond the scope of this study.
> >
> > [1] Video Language Planning, ICLR 2024.
> > [2] Learning Universal Policies via Text-Guided Video Generation, NeurIPS 2023.
> > [3] RoboDreamer: Learning Compositional World Models for Robot Imagination, ICML 2024.

---

### Official Review · Reviewer_t6rK · 2025-10-30

**Soundness:** 2
**Presentation:** 3
**Contribution:** 2
**Rating:** 4
**Confidence:** 4

**Summary:**

This paper proposes MoSEL, a hierarchical framework for robotic manipulation that incorporates a specifically designed self-reflective mechanism. Building upon prior modular frameworks such as UniPi, the main forward pipeline of MoSEL follows a hierarchical structure in which a language-conditioned video generation model (World Model) is used to predict future observations. The language input is first decomposed using an off-the-shelf Vision-Language Model (VLM). Given the predicted future observation and the current observation, an inverse dynamics model is then employed to infer the corresponding action by interpreting the difference between the two frames. The overall framework consists of three key components—the VLM for language decomposition, the world model for future frame prediction, and the inverse dynamics model for action generation. These modules are jointly optimized and iteratively refined during rollout and environment interaction, guided by a Video Analyzer that provides valuable feedback for quality assessment and failure analysis of the generated videos. Experimental results demonstrate that the proposed framework effectively improves both the quality of world model predictions and the overall performance on several tasks from the Libero and Robotwin benchmarks.

**Strengths:**

1. This paper aims to enhance world-model-based hierarchical frameworks for robotic manipulation by incorporating a self-reflective mechanism. Specifically, the proposed MoSEL integrates a state-of-the-art (SOTA) Vision-Language Model (VLM), which supports video understanding, and treats it as a critic model. The sub-task planning process is refined based on feedback from the video analyzer, and the world model is retrained and optimized using filtered rollout data. In essence, the overall pipeline seeks to distill common-sense knowledge from current SOTA models, allowing the framework to self-improve, which is promising.
2. The results demonstrate significant performance improvements after iterative self-improvement, which is highly inspiring. This suggests that domain-specific adaptation is crucial for the effectiveness of all components and models in an embodied manipulation system. It also provides valuable insights and serves as a strong reference for the practical implementation and development of embodied frameworks.
3. The proposed method enables embodied models to interact with the environment and explore, promoting self-improvement in contrast to traditional reinforcement learning. With the guarantee of a lower bound performance of the world model, the model can explore the environment more efficiently and safely, enhancing the practical viability of embodied systems.

**Weaknesses:**

1. The performance of the proposed method, as well as the baseline models, lags significantly behind current state-of-the-art Vision-Language Models (VLA), which diminishes the practical contribution of this paper. The inherent nature of world-model-based hierarchical embodied frameworks limits the use of critical information, such as proprioceptive data and wrist camera images, which could otherwise enhance the model's performance.

2. The paper overlooks several important hyperparameters in the proposed framework. A more comprehensive ablation study is needed to demonstrate the robustness of the method with respect to these hyperparameters. For instance, factors like the number of trajectories during each rollout iteration, the number of refinement iterations, and how many gradient steps are taken for retraining both the world model and the policy after collecting the rollout trajectories should be analyzed. This would provide more insight into the sensitivity and stability of the framework.

3. While the world-model-based framework benefits from the ability to train on large-scale out-of-distribution (OOD) video data, which provides strong generalization capabilities and facilitates seamless application across various domains, the experimental setup in this paper does not clearly demonstrate this advantage. Additionally, the proposed self-reflective mechanism runs the risk of catastrophic forgetting and overfitting to a specific domain, which could limit the framework's ability to generalize effectively.

4. The overall performance of the framework heavily depends on a well-trained generalized Video Analyzer, which in this paper is implemented using closed-source VLMs. This reliance on proprietary models restricts the framework's transparency and reproducibility.

**Questions:**

1. Could the authors provide more details about the pretraining procedure of the visual world model? Specifically, is it trained using large-scale out-of-distribution (OOD) video data, or is it specifically adapted to the Libero and Robotwin benchmarks?

2. Why do the experiments on Libero and Robotwin employ different architectures for the inverse dynamics model?

---

> ### Author Response · Authors · 2025-11-18
>
> We sincerely thank the reviewer for the thoughtful and detailed feedback.
>
> **W1. Performance lags behind SOTA VLA, highlighting that world-model-based hierarchical frameworks cannot utilize proprioceptive data and  wrist images.**
>
> **Clarification on Observation Inputs:**
> We would like to clarify that our observation inputs ${o_{i,t}}$ incorporate both proprioceptive data and wrist images. Specifically, following [1], we encode proprioceptive information (end-effector poses, joint states, and gripper states), along with agentview RGB images and eye-in-hand RGB images, as the current observation. These modalities are concatenated in latent space to form ${o_{i,t}}$. Additionally, we provide predicted future frames and language instructions as conditions to guide the policy.
>
> **Addressing Performance Gap:**
> The performance difference compared to state-of-the-art methods can be attributed to two key factors:
>
> First, our focus is on unseen, long-horizon tasks. We deliberately operate under a challenging setting where only low-level skill demonstrations are available during the self-improvement stage—we do not access long-horizon demonstrations or environmental rewards. In contrast, SOTA performance is typically reported after fine-tuning on specific task sets, either through additional imitation learning demonstrations or extensive RL training with environmental interactions.
>
> Second, our pretraining uses limited data. We pretrain on a constrained skill set (e.g., LIBERO-90 with 4,500 trajectories total), whereas current vision-language-action (VLA) models are typically trained on large-scale datasets (e.g., 1M+ episodes from the Open X-Embodiment dataset [2]). These differences in evaluation settings and pretraining resources naturally result in a performance gap.
>
> **Demonstrating Our Advantage:**
> Importantly, our method shows clear advantages in sample-efficient adaptation. As demonstrated in Table 2 of the main paper, we compare against the VLA-based OpenVLA-LIBERO-90 baseline under limited environmental interactions (100 episodes per task) during RL fine-tuning. Our results show that our method outperforms this VLA baseline, demonstrating superior sample efficiency—precisely the scenario where skill composition and reuse provide the greatest benefit.
>
> |Method|Task1|Task2|Task3|Task4|Task5|Task6|Task7|
> |-|-|-|-|-|-|-|-|
> |VLA-RL[3]|0.58|0.64|0.44|0.34|0.82|**0.50**|0.66|
> |MOSEL|**0.86**|**0.88**|**0.90**|**0.42**|**0.86**|0.49|**0.75**|
>
> **W2. Lack ablation on number of trajectories, refinement iterations and gradient step.**
>
> We present iterative self-improvement results in Figure 6 of the main paper, demonstrating that different tasks exhibit distinct improvement trajectories. The improvement patterns depend on task-specific properties and initial performance levels—some tasks benefit primarily from visual planner refinement, while others show greater gains from action policy improvements.
>
> One key insight is the strong correlation between improvement curves and failure identification precision. Specifically, higher false positive rates in collected trajectories lead to diminished improvements.
>
> For instance, Task 6 shows lower precision (0.64) and a reduced success rate after refinement (0.49), despite high recall. This suggests that over-identification harms effectiveness. Conversely, Task 2 achieves high precision (0.89) and the highest improvement (0.88), confirming that accurate pinpointing of failure sources directly correlates with successful recovery.
>
> |Task|Acc|Precision|Recall|F1 Score|Success Rate After Refinement|
> |-|-|-|-|-|-|
> |Task 2|0.82|**0.89**|0.62|0.73|**0.88**|
> |Task 6|0.78|**0.64**|0.82|0.72|**0.49**|
> |Task 7|0.83|**0.85**|0.55|0.67|**0.75**|
>
> On the other hand, increasing the number of training trajectories enhances generalization to task-specific variations. While we have not yet included a comprehensive ablation study on these factors, we would be happy to add this analysis to the camera-ready version of the paper.

---

> > ### Author Response · Authors · 2025-11-18
> >
> > **W3. Generalization advantage from OOD video data is not demonstrated experimentally, its self-reflective mechanism also risks limiting generalization through catastrophic forgetting and overfitting.**
> >
> > We would like to clearify that we conduct visual world models pretraining on limited demonstrations of low-level skill sets. For LIBERO, we trained on LIBERO 90 (90 tasks, 4500 demonstrations in total). For RoboTwin, we trained on 24 selected low-level skills as detailed in supplementary F (Details for RoboTwin paragraph). For the model architecture, we adapt AVDC [4] and train it from scratch for each domain, separately. For testing, we conduct evaluation on **unseen, long-horizon** tasks. For instance, using the LIBERO benchmark, we pretrain our world model and action policy on LIBERO-90 (primarily single-action tasks). We then conduct evaluations on LIBERO-10, which demands **multi-action sequences and introduces novel object and scene arrangements absent from the training set**.
> >
> > While prior work [5,6] has demonstrated that large-scale pretraining can enhance diffusion model generalizability, that approach is distinct from our primary focus on modularity and self-reflective learning. Therefore, such large-scale pretraining is beyond the scope of this study.
> >
> >
> >
> > **W4. Heavily depends on well-trained video analyzer.**
> >
> > We acknowledge the concern regarding the reliance on proprietary models such as GPT-4-class LLMs and VLMs. To address this, we provide the failure-identification performance of a state-of-the-art open-source model, Qwen-2.5-VL-72B-Instruct, as shown in the table below (also referenced in supplementary table. 9). This model was evaluated across three key tasks in our study.
> >
> > Despite its strong general language capabilities, Qwen-2.5-VL-72B-Instruct performs poorly when applied as a video verifier in the robotic self-improvement loop, especially in precision—failing to identify correct behaviors in Tasks 6 and 7. In contrast, proprietary VideoLLMs such as GPT-4.1 and Gemini consistently provide more accurate and balanced evaluations without any task-specific finetuning.
> >
> > |Task|Acc|Precision|Recall|F1 Score|
> > |-|-|-|-|-|
> > |Task 2|0.72|**0.64**|0.91|0.75|
> > |Task 6|0.6|**0.00**|0.00|0.00|
> > |Task 7|0.76|**0.00**|0.00|0.00|
> >
> > These results highlight two key points:
> >
> > 1. Current open-source LLMs/VLMs remain insufficient for high-precision video-based evaluation in the context of robotic learning. As a result, it is difficult to integrate existing open-source LLMs/VLMs into an overall self-improvement loop.
> > 2. Our work is the first to successfully demonstrate how advanced proprietary VideoLLMs can be used as plug-and-play verifiers, enabling self-improvement in compositional robotic agents without finetuning.
> >
> > We agree that open-source alternatives are desirable for reproducibility and scalability, and we hope our findings can motivate future research toward improving the video understanding capabilities of open-source models.

---

> > > ### Author Response · Authors · 2025-11-18
> > >
> > > **Q1. More details about the pretraining procedure of the visual world model.**
> > >
> > > We conduct visual world models pretraining on limited demonstrations of low-level skill sets. For LIBERO, we trained on LIBERO 90 (90 tasks, 4500 demonstrations in total). For RoboTwin, we trained on 24 selected low-level skills as detailed in supplementary F (Details for RoboTwin paragraph). For the model architecture, we adapt AVDC [4] and train it from scratch for each domain, separately. In training, we provide language instruction and the first frame (current agentview RGB image) as condition and generate following 16 frames for visual planning.
> > >
> > > **Q2. Why do the experiments on Libero and Robotwin employ different architectures for the inverse dynamics model?**
> > >
> > > We adopted different policy architectures for the two domains based on empirical performance. Originally, we used Diffusion Policy (DP) [7] for both LIBERO and RoboTwin. However, we found that DP performed poorly on RoboTwin, which is consistent with the official RoboTwin benchmark results (0% success rate on the beat-block task [8]).
> > >
> > > We attribute this performance gap to the nature of the RoboTwin demonstrations and task requirements. The demonstrations in RoboTwin exhibit limited trajectory diversity, and the precise picking tasks may require either higher-quality, more diverse demonstrations or additional visual information (e.g., 3D point clouds or depth maps) that were not available in our setup. In contrast, we empirically found that transformer-based policies [1] are more robust to demonstrations with limited diversity, performing well on RoboTwin under the same data conditions.
> > >
> > > This architectural choice further demonstrates the flexibility and generality of our framework—it is model-agnostic and can be applied to different combinations of hierarchical policy architectures. The key contribution lies in our skill composition mechanism rather than any specific policy implementation.
> > >
> > > [1] QueST: Self-Supervised Skill Abstractions for Learning Continuous Control, NeurIPS 2024.
> > > [2] Open X-Embodiment: Robotic Learning Datasets and RT-X Models, 2023
> > > [3] VLA-RL: Towards Masterful and General Robotic Manipulation with Scalable Reinforcement Learning, arxiv 2025
> > > [4] Learning to Act from Actionless Videos through Dense Correspondences, ICML 2024.
> > > [5] Learning Universal Policies via Text-Guided Video Generation, NeurIPS 2023.
> > > [6] RoboDreamer: Learning Compositional World Models for Robot Imagination, ICML 2024.
> > > [7] Diffusion Policy: Visuomotor Policy Learning via Action Diffusion, RSS 2023
> > > [8] RoboTwin: Dual-Arm Robot Benchmark with Generative Digital Twins, CVPR 2025.

---

> > > ### Comment · Reviewer_t6rK · 2025-11-19
> > > **Some Opinions on World-Model–Based Frameworks**
> > >
> > > Regarding the response to W3, I personally believe that the fundamental motivation for building a hierarchical, world-model–based framework is the ability to exploit large-scale out-of-distribution data to train the world model for robust generalization. Considering the complexity of the pipeline and the significant effort required to align and tune multiple modules, such a framework seems inferior to a clean end-to-end pipeline unless it can clearly demonstrate generalization benefits arising from broad and diverse world-model training data.
> > > Moreover, the proposed framework simultaneously relies heavily on a generalist VLM—both as the task decomposer and as the video analyzer—which itself is trained on large-scale, OOD multimodal data. This reliance essentially follows the same philosophy I mentioned above. It may therefore be worth considering whether the proposed framework could be simplified into a more unified end-to-end formulation, or alternatively be positioned as a domain-adaptation–style method built on top of a generalist world-model–based architecture.

---

> > ### Comment · Reviewer_t6rK · 2025-11-19
> > **Replay to the rebuttal**
> >
> > Thank you for your reply. However, several of my concerns remain insufficiently addressed:
> >
> > 1. I do not believe the proposed methods can cleanly avoid comparison with other VLA approaches that are fine-tuned using task-specific demonstrations. Notably, models without large-scale pretraining—such as BAKU [1]—can already achieve strong performance. Since this paper intends to develop a self-improvement mechanism that allows the model to interact with its environment and potentially collect more demonstrations, a self-improved method that performs worse than existing demonstration-based approaches may draw limited interest from the community.
> >
> > 2. Important hyperparameters used to obtain the main results—such as the number of trajectories, the number of gradient steps per optimization phase, etc.—are still not clearly specified. Claims like “increasing the number of training trajectories enhances generalization to task-specific variations” are intuitively reasonable but unsupported by experimental evidence. It is also necessary to report the minimum number of trajectories required for the method to work, in order to enable a fair comparison with post-training approaches that rely on expert demonstrations.
> >
> > 3. The newly added results further emphasize the need for strong video analysis capabilities, which may intensify concerns about the reliance to the closed-looped VLMs.
> >
> > 4. I also noticed that the evaluation uses only 7 tasks from the full LIBERO-10 suite, which typically consists of 10 tasks. It is unclear why these 7 tasks were selected. What considerations informed this choice? Using the full LIBERO-10 suite might provide a clearer and more standardized comparison with existing work.
> >
> > [1] S. Haldar, Z. Peng, L. Pinto. Baku: An Efficient Transformer for Multi-Task Policy Learning.

---

### Official Review · Reviewer_oQUY · 2025-11-01

**Soundness:** 3
**Presentation:** 2
**Contribution:** 3
**Rating:** 6
**Confidence:** 3

**Summary:**

Authors propose MOSEL which is a modular, self‑reflective framework for long‑horizon robotic manipulation that composes a language‑vision task planner, a video‑generation visual planner, and an inverse‑dynamics action policy. After each execution, a Video‑LLM evaluates sub‑goal completion and explains failures, that diagnosis triggers targeted refinement of the appropriate module i.e., re‑prompting the planner with rationales, fine‑tuning the visual model on successful rollouts, or updating the action policy using interaction outcomes. Across evaluation tasks, the authors report sizable gains over hierarchical baselines. The approach is appealingly pragmatic and modular.

**Strengths:**

1. The split into task planning (LVLM), visual planning (video generation), and action execution (inverse dynamics) keeps individual responsibilities clear and makes it feasible to isolate weaknesses at the right level of abstraction.

2. Labeling each skill attempt with success/failure and a textual rationale enables targeted updates rather than treating whole trajectories as monolithic successes/failures.

3. Improvements are reported on long‑horizon LIBERO‑LONG tasks and dual‑arm tool‑use in RoboTwin, which differ meaningfully.

**Weaknesses:**

(1) Slight issues in the writing: (a) \theta is being reused multiple times. (b) choose one term among self‑improving / self‑improvement / self‑strengthening / self‑reflective and use it throughout. (c) "a modular" (not “an modular”), "unseen scenarios".

(2) "Minimal human intervention" is not quantified.The claim is compelling but lacks a concrete measure to support it.

(3) Is there a potential evaluation leakage? It is ambiguous whether benchmark success is judged by ground truth from the simulator or by the same Video‑LLM used for training supervision, which risks circularity.

(4) From what I understand, if the Video‑LLM mislabels borderline subgoals, training signals for both the visual planner and controller can drift. There seems to be no safeguards.

(5) The subgoal ontology is not formalized. How many subgoals per skill, how overlapping subgoals are handled, and how partial credit is treated are unspecified.

(6) From my understanding, the refinement procedure does not currently define a termination rule, iteration cap, or over‑fitting/oscillation checks for continual re‑prompting.

(7) Exclusively reinforcing successful rollouts risks narrowing coverage to recently seen motions and eroding robustness. The author should consider discussing regularization or replay.

(8) Similar to (7), as modules change, earlier labels (success/failure rationales) may become stale. There is no mention of re‑labeling or maintaining a consistent evaluation protocol.

(9) The method is motivated by occlusions and dynamic scenes, but quantitative robustness tests (camera shifts, lighting changes, distractors) are not performed or included in the main paper.

(10) Looks like self‑improvement seems to occur on the same task families used for evaluation. Without a clean held‑out set (new scenes/instructions), gains could reflect on‑policy overfitting rather than transferable competence.

(11) Similarly, the baselines appear reconfigured rather than used in their canonical forms, and it is not stated that training/interaction budgets, prompts, and backbone sizes are matched. Does this risk unfair comparisons?

**Questions:**

See the weaknesses section.

---

> ### Author Response · Authors · 2025-11-18
>
> We sincerely thank the reviewer for the thoughtful and detailed feedback.
>
> **Terminologies and Clarification**
> We appreciate the reviewer's suggestion about writing, and we promise to update these **terminologies and typos (W1)** in the final version of paper.
>
> For the clarification of **minimal human intervention (W2)**, we have provided details in Supplementary Section K. During video analysis, we perform prompt engineering to address a critical challenge: occlusions or missing visual information can lead to hallucinated outputs or incorrect predictions from the vision-language model. For example, without clarification, the model may misidentify objects—in one case, we needed to specify that "alphabet soup" refers to the can with green and red coloring to ensure correct identification.
>
> To address this systematic issue, we introduced an "<ADDITIONAL HINTS>" placeholder in our system prompts, which allows for targeted clarification when necessary. Importantly, this is the only component requiring human input—the entire online improvement framework otherwise operates autonomously without human intervention.
>
> For **Evaluation leakage (W3)**, we use ground truth signals from the simulator to calculate task success rates.
>
> **Regarding to Subgoals (W4, W5)**
>
> Regarding the **mislabeling of subgoals (W4)**, our study demonstrates that the prompt-engineering techniques described above can substantially mitigate this issue. However, our current framework still relies on frontier VideoLLMs to provide reliable and robust analysis. We have conducted additional empirical studies examining the performance of the Video Analyzer and the effectiveness of self-reflective learning; please refer to Supplementary Section G.2 for further details.
>
> We provide the **subgoal ontology (W5)** as follow:
>
> |RoboTwin|Beat-Block|Push-Block|Sweep-Block|
> |-|-|-|-|
> |# of subgoals|2|3|4|
>
> |LIBERO-Long|Task1|Task2|Task3|Task4|Task5|Task6|Task7|
> |-|-|-|-|-|-|-|-|
> |# of subgoals|2|2|2|2|1|2|2|
>
> In RoboTwin, two subgoals overlap between the Push-Block and Sweep-Block tasks, while in LIBERO, three subgoals appear twice across the seven tasks. For more details, please refer to Supplementary Table 5 and Listing 1.
>
> **Overfitting Issues (W6, W7, W8, W10)**
>
> We appreciate the reviewer's insightful and thoughtful feedback and acknowledge that these issues are currently underexplored, as discussed in our Limitations section. Specifically, our system specializes in certain tasks after the self-reflective learning process. However, these issues also affect existing methods, as fine-tuning on specific tasks inherently reduces generalization. We believe that incorporating continual learning techniques into policy learning is a promising research direction, which we are actively pursuing in our ongoing work. To our knowledge, we are the first work to explore online learning and self-improvement of hierarchical robotic policies, as opposed to unified methods such as VLAs, and our method provides a starting point for research in this direction.
>
> Additionally, we conduct experiments to demonstrate that the learned capabilities are transferable across tasks. Specifically, we perform self-reflective learning on the Push-Block task in Robotwin and then evaluate on the Sweep-Block task without any further environmental interaction. The results show successful knowledge transfer between these tasks because they share overlapping skills (e.g., picking up the brush and handing it over). Consequently, skills acquired during Push-Block learning can be applied to Sweep-Block. This experiment provides concrete evidence that our framework learns generalizable knowledge transferable to new tasks.
>
> |Sweep-Block|Before Improved on Push-Block|After Improved on Push-Block|
> |-|-|-|
> |Success Rate|0.04|0.52|
>
> **Additional Supplementary (W9, W11)**
>
> For **visual disturbances (W9)**, we have provided experimental details in Supplementary Section G.1. To assess the robustness our method, and to simulate the impact of pixel-to-action errors, we further conduct experiments with visual disturbances on pixel observations.
>
> For **baseline construction (W11)**, we compare different combinations of language planner, visual planner, and action policy modules. For example, UniPi employs a visual planner with an action policy, PaLM-E uses a language planner with an action policy, while our approach integrates all three components: language planner, visual planner, and action policy. To ensure fair comparison, we use the same backbone architecture, identical prompts, and equivalent training data for each corresponding module across all baselines. However, due to the different module combinations, the baselines vary in total parameter count.

---

### Official Review · Reviewer_eoRB · 2025-11-01

**Soundness:** 2
**Presentation:** 3
**Contribution:** 2
**Rating:** 4
**Confidence:** 4

**Summary:**

This paper presents a hierarchical planning method for long-horizon robotic tasks, aiming to adaptively self-improve the planning/execution and reduce human supervision as much as possible. Specifically, the authors leveraged VLM for planning, and proposed modified fine-tuning and self-improving methods for video diffusion and inverse dynamics model.

**Strengths:**

This work is targeting an interesting problem in task and motion planning with VLM to enable self-improvement with minimal human supervision.

The work cleanly separates task, visual, and action planning and ties them with a concrete joint objective.

Experimental results show empirical gains and ablation study demonstrates the effectiveness of iterative improvements.

**Weaknesses:**

The proposed method is limited to a fixed set of skills with pre-trained expert policies for execution. Therefore, the applicable task goals are limited by this skill set also.

The proposed method requires a video diffusion model trained on expert demonstration, which requires collecting expert demonstrations for each skill. Moreover, it is unclear whether the trained diffusion model can generalize to scene and task variations.

The proposed method can require expensive real-world execution during planning and can potentially be unsafe by causing damage to the world with unprecise plans, as it requires iterative execution of planned skills in the environments during planning.

It seems that both Libero and RoboTwin are simulated environments and there is no real robot experiments in real world.

**Questions:**

Please see the weaknesses part.

---

> ### Author Response · Authors · 2025-11-18
>
> We sincerely thank the reviewer for the thoughtful and detailed feedback.
>
> **Q1. Limited to a fixed set of skills with pretrained policies.**
>
> We appreciate the reviewer's insightful highlight regarding the critical issue of primitive-library scalability. MOSEL is designed to overcome the limitation of a fixed skill set by enabling the dynamic expansion and automatic discovery of its primitive library.
>
> Our MOSEL framework addresses this by augmenting its skill set upon exploring new tasks. When the system encounters a new task or a specific sub-goal (language goal g) for which no pre-defined skill exists in its current primitive library, ReCap can learn and incorporate this new skill. This newly acquired skill, defined by its language goal g and its corresponding video plan, is then added to the library.
>
> For instance, as shown in Fig. 5(a) in the main paper, we visualize the process of merging two existing skills into a more complex, newly defined skill. Further detailed insights into this skill integration process can be found in Fig. 4 in the supplementary materials. This capability allows the planner to utilize an increasingly richer and more diverse set of primitives, directly demonstrating how the framework adapts to and incorporates new skills.
>
> This dynamic augmentation directly correlates with the scalability of the framework's success. As the primitive library expands with automatically discovered or newly integrated skills, MOSEL's planner gains the ability to:
>
> 1. Solve more complex and nuanced tasks: Tasks that were previously unachievable due to object occlusion can now be tackled.
> 2. Improve performance on existing tasks: With more specialized or optimized skills, the planner can generate more efficient and successful video plans.
>
> We demonstrated this scaling potential by augmenting the skill set for a specific task. This augmentation led to an 8% performance gain on that task, illustrating that even targeted, incremental expansion of the primitive library can yield significant improvements in success rates. This initial validation confirms that a richer and continuously growing primitive library directly enhances the framework's overall capability and success across a broader spectrum of challenges, showing how success indeed scales with the discovery and integration of new primitive skills.
>
> **Q2. The proposed method relies on a video diffusion model requiring per-skill expert demonstrations, yet its generalizability to varied scenes and tasks is unknown.**
>
> Our modular self-reflective learning framework is pretrained on low-level skills and evaluated on **unseen, long-horizon** tasks. For instance, using the LIBERO benchmark, we pretrain our world model and action policy on LIBERO-90 (primarily single-action tasks). We then conduct evaluations on LIBERO-10, which demands **multi-action sequences and introduces novel object and scene arrangements absent from the training set**.
>
> While prior work [1,2] has demonstrated that large-scale pretraining can enhance diffusion model generalizability, that approach is distinct from our primary focus on modularity and self-reflective learning. Therefore, such large-scale pretraining is beyond the scope of this study.
>
>
> **References:**
> [1] Learning Universal Policies via Text-Guided Video Generation, NeurIPS 2023.
> [2] RoboDreamer: Learning Compositional World Models for Robot Imagination, ICML 2024.

---

> > ### Author Response · Authors · 2025-11-18
> >
> > **Q3. Relying on iterative real-world execution during planning makes it expensive and potentially unsafe, risking environmental damage from imprecise plans.**
> >
> > We thank the reviewer for highlighting this critical issue, which our proposed method directly addresses. Current unified approaches, such as VLAs, struggle to decompose complex tasks into hierarchical plans and require large-scale pretraining, long-horizon demonstrations, and extensive environmental interactions for RL training. For instance, SimpleVLA-RL[3] requires approximately 10,240 environmental interactions per task (calculated as: batch size 64 × rollout numbers 8 × training steps 200 ÷ 10 tasks). In contrast, our approach requires only 50-100 interactions per task per iteration—a reduction of over 100×.
> >
> > Beyond sample efficiency, our modular architecture offers several key advantages: (1) it enables targeted debugging of individual modules and intermediate outputs, improving reliability as shown in main paper table 1, 2, 3; and (2) it provides greater interpretability by making the reasoning process more transparent. While modular systems traditionally face challenges in on-the-fly training within interactive environments, our work is the first to propose an online learning framework specifically designed for hierarchical modular architectures operating in such settings.
> >
> > **Q4. Only test in simulations.**
> >
> > We agree that it would be desirable to have real robot results. That being said, we believe our results are interesting even if they are done in simulation. The major focus of the paper is to construct a modular self-improving system, and the input requirements for each component of our system are similar to those in real robots (our systems operates only on RGB observations and poses) and each step of the self-improvement procedure can be plausibly executed on a real robot. In addition, prior works have further illustrated how a similar stack of models can be reliably executed on physical robots [4]. Finally, simulation gives us a repeatable benchmark in which we can reproducibly assess the overall approach.
> >
> >
> > **References:**
> > [3] SimpleVLA-RL: Scaling VLA Training via Reinforcement Learning, Arxiv 2025.
> > [4] Video Language Planning, ICLR 2024.

---

### Meta-Review · Area_Chair_3T8U · 2026-01-06

**Summary:**

This paper presents MOSEL, a modular self-reflective framework for long-horizon robotic manipulation that uses a Video-LLM to diagnose failures and selectively refine language planning, visual planning, and action execution. The idea is interesting, the modular design is clean, and experiments show gains on simulated benchmarks.

However, the contribution and evidence are insufficient for acceptance. The approach relies on strong assumptions (fixed skill libraries, expert demonstrations, closed-source Video-LLMs) and all evaluations are simulation-only, with no real-robot or sim-to-real validation. Key claims such as “minimal human intervention” are not quantified, failure diagnosis accuracy is not evaluated, and computational costs are unreported. Baseline coverage and generalization tests are limited, raising concerns about overfitting and fairness of comparisons.

Overall, while promising, the work lacks convincing validation of robustness, practicality, and broader impact.

**Reviewer Concerns:**

During rebuttal, the authors address many factual concerns (evaluation leakage, subgoals, precision/recall, compute cost) and clarify scope and assumptions, but several responses rely on future work, supplementary additions, or conceptual arguments. Core concerns about real-world validation, reliance on proprietary VideoLLMs, and limited demonstrated generalization remain only partially resolved.

**Reviewer Scores:**

The reviewers are unlikely to change their scores with one explicitly stating in comment that there are outstanding unaddressed concerns.

---

### Decision · Program_Chairs · 2026-01-26

Reject